# Enabling direct-growth route for highly efficient ethanol upgrading to long-chain alcohols in aqueous phase

Juwen Gu [1,6], Wanbing Gong[2,6], Qian Zhang[1,3,4,6], Ran Long [2], Jun Ma [2], Xinyu Wang[2], Jiawei Li[2], Jiayi Li[2], Yujian Fan[1], Xinqi Zheng[1], Songbai Qiu [1,3,4] ✉, Tiejun Wang [1,3,4] ✉ & Yujie Xiong [2,5] ✉

Upgrading ethanol to long-chain alcohols (LAS, $C_{6+}OH$) offers an attractive and sustainable approach to carbon neutrality. Yet it remains a grand challenge to achieve efficient carbon chain propagation, particularly with noble metal-free catalysts in aqueous phase, toward LAS production. Here we report an unconventional but effective strategy for designing highly efficient catalysts for ethanol upgrading to LAS, in which Ni catalytic sites are controllably exposed on surface through sulfur modification. The optimal catalyst exhibits the performance well exceeding previous reports, achieving ultrahigh LAS selectivity (15.2% $C_6OH$ and 59.0% $C_{8+}OH$) at nearly complete ethanol conversion (99.1%). Our in situ characterizations, together with theoretical simulation, reveal that the selectively exposed Ni sites which offer strong adsorption for aldehydes but are inert for side reactions can effectively stabilize and enrich aldehyde intermediates, dramatically improving direct-growth probability toward LAS production. This work opens a new paradigm for designing high-performance non-noble metal catalysts for upgrading aqueous EtOH to LAS.

Long-chain alcohols are important and valuable building blocks for the massive production of high-quality biofuels and fine chemicals such as plasticizers, thickening agents, lubricants and detergents in modern chemical industry[1-4]. Nowadays, the long-chain alcohols are dominantly synthesized industrially via the traditional oxo synthesis, which requires expensive organometallic catalysts, tedious synthetic procedures and usage of petroleum-derived olefins as feedstocks[5]. Yet in recent years, for achieving the goal of carbon neutrality, the sustainable production of LAS from green carbon resources such as biomass-derived ethanol (EtOH) has attracted increasing attention, because of

its huge potential in reducing the dependence on fossil resources and the environmental impact. As an important industrial process, the renowned Guerbet condensation, occurring via highly reactive aldehyde intermediates, is considered as an atom- and step-economical approach to increase the carbon numbers of short-chain alcohols[6-8]. In general, the Guerbet condensation of EtOH proceeds in a series of tandem reactions, which involves alcohol dehydrogenation to acetaldehydes and their aldolization reaction followed by dehydration to conjugated alkenal and subsequent hydrogenation to n-butanol and LAS (Fig. 1a).

[1]School of Chemical Engineering and Light Industry, Guangdong University of Technology, Guangzhou 510006, China. [2]Hefei National Research Center for Physical Sciences at the Microscale, Collaborative Innovative Center of Chemistry for Energy Materials (iChEM), School of Chemistry and Materials Science, National Synchrotron Radiation Laboratory, School of Nuclear Science and Technology, University of Science and Technology of China, Hefei 230026 Anhui, China. [3]Guangdong Provincial Key Laboratory of Plant Resources Biorefinery, Guangzhou 510006, China. [4]Guangzhou Key Laboratory of Clean Transportation Energy and Chemistry, Guangzhou 510006, China. [5]Suzhou Institute for Advanced Research, Nano Science and Technology Institute, University of Science and Technology of China, Suzhou 215123, China. [6]These authors contributed equally: Juwen Gu, Wanbing Gong, Qian Zhang. ✉e-mail: qiusb@gdut.edu.cn; tjwang@gdut.edu.cn; yjxiong@ustc.edu.cn

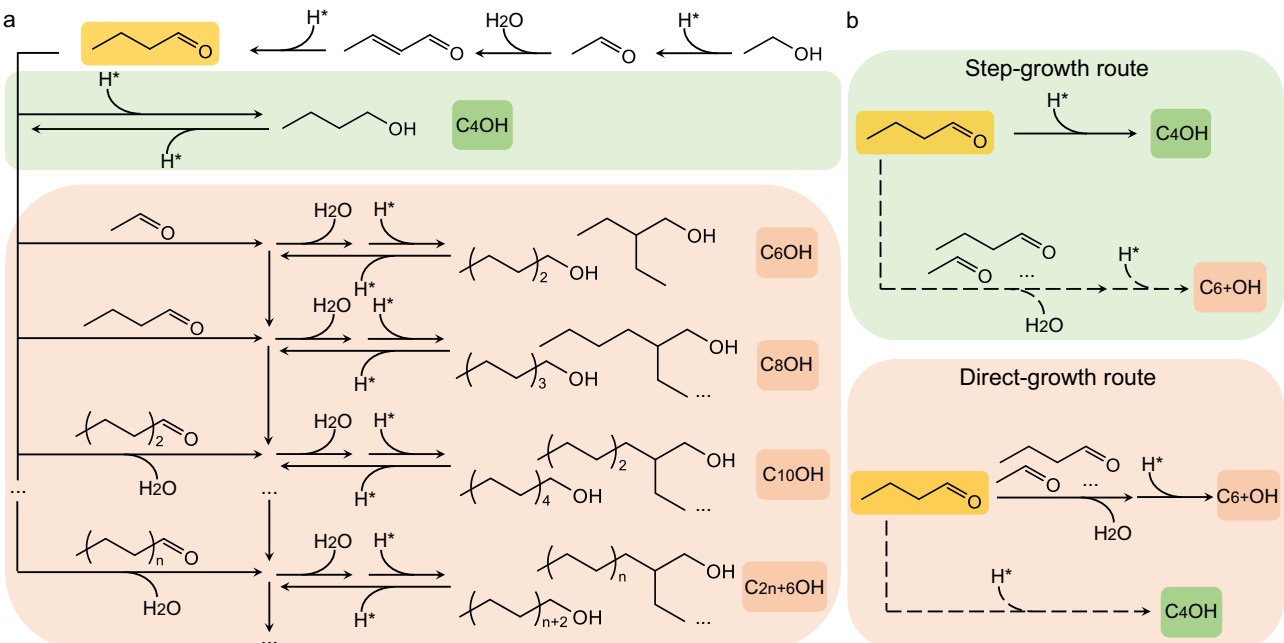

**Fig. 1 | Schematic illustration of upgrading ethanol to long-chain alcohols. a** Reaction network of Guerbet condensation. **b** Step-growth route for the production of LAS by repeating the processes of Guerbet condensation (upper part), and direct-growth route for the production of LAS by condensation of generated aldehydes directly (lower part).

To trigger the desired Guerbet condensation, the rate-limiting step (RLS) of alcohol dehydrogenation and competitive side reactions should be taken into account when designing catalytic systems. To this end, the introduction of transition metals (e.g., Cu, Ni and Pd) into heterogeneous catalysts can facilitate the molecular adsorption and dehydrogenation of starting alcohol, consequently promoting EtOH coupling at milder temperatures. However, pioneering works primarily focused on catalytic coupling of ethanol to *n*-butanol with high selectivity and conversion through modulating active metal sites and basic/acidic sites[9–12]. Frustratingly, the LAS selectivity is still extremely low due to lack of high-performance catalysts in breaking the restriction of the classic step-growth model (upper part of Fig. 1b) which represents a progressive decline in the alcohol selectivity from *n*-butanol to the alcohols with longer chain[13–17]. Given that the dehydrogenation of *n*-butanol is more difficult than that of EtOH[15], it should be an efficient pathway for improving the LAS selectivity by directly shifting the generated *n*-butanal to next aldol condensation rather than complete hydrogenation (lower part of Fig. 1b). Zhu et al. have demonstrated that enriching aldehyde intermediates can improve the direct-growth probability, increasing the LAS selectivity up to 67.7% with 55.6% of EtOH conversion over a biomass-derived Ni/apatite catalyst[18]. However, the Ni loading was limited within 1 wt% because excessive Ni species could suppress chain propagation by covering condensation sites and cause severe C–C bond cleavage, thereby limiting overall reaction efficiency and practical applications. As such, a new design paradigm, which can stabilize and accumulate more aldehyde intermediates on catalytically active sites for accelerating EtOH coupling toward LAS, is highly desired yet challenging.

To open up such a new paradigm, we look into an unconventional strategy of using sulfur modification on metal catalysts. For most metal catalysts, sulfur is traditionally recognized as a severe poison because sulfur bonds strongly to metal surface. Only a few reports suggested that the promotional effect both on the activity and selectivity could be implemented by altering the adsorption properties of active metal sites via controllable sulfur modification, in some reactions such as Fischer–Tropsch synthesis, hydrogenation and

dehydrogenation[19–24]. In parallel, our previous works have demonstrated that the Sn- or N-modified Ni catalysts efficiently catalyze aqueous EtOH coupling to C₄₊OH by weakening the adsorption of aldehyde intermediates on the surface Ni sites; however, the corresponding alcohol distribution still succumbs to the step-growth model, giving -50% of *n*-butanol as the major product[13,25–27]. Against this background, the strategy of tuning intermediate adsorption faces a dilemma that the strong adsorption of aldehyde intermediates on Ni sites can enrich the intermediates to ensure the following condensation, but the key aldehydes adsorbed strongly on the Ni surface have to be stabilized preferentially to avoid excessive C–C bond cleavage toward undesired small-molecule byproducts. To overcome the dilemma, we propose that it should be a feasible way to highly selective LAS production by fabricating unique Ni surface with sulfur modification, which may offer strong adsorption sites for aldehydes but be inert for side reactions. Nevertheless, it remains challenging to achieve the controllable exposure of Ni sites and their modification with sulfur species on catalyst surface.

Here, we report the design of simple and efficient Ni catalysts that are encapsulated in graphitized carbon and modified with sulfur (namely, Ni@C-Sₓ, x stands for S/Ni molar ratio), which is realized through one-step carbonization of nickel–organic complex gels containing small amounts of sulfur. Enabled by this design, incorporating an appropriate amount of sulfur into the catalysts can not only accelerate the EtOH conversion to LAS, but also break limitation on carbon chain propagation involved in the current step-growth model. Optimizing the sulfur content, the Ni@C-S₁/₃₀ catalyst achieves unprecedentedly high performance with up to 74.2% of LAS selectivity (15.2% of C₆OH and 59.0% of C₈₊OH) at an EtOH conversion of 99.1% within 12 h. Our in situ characterization and density functional theory (DFT) calculation results reveal that the sulfur atoms forming strong Ni–S bonds can selectively block the active Ni sites on surface from cleavage reaction pathways through the effect of steric hindrance, while retaining most of the strong absorption sites for enriching aldehyde intermediates. We further demonstrate that the surface sulfur modification is a generic strategy which allows for flexibility in the selection of various sulfur precursors and adding methods.

## Results and discussion

### Structural characterizations

The Ni@C-$S_x$ catalysts were synthesized by one-step carbonization of nickel organic complex gels containing small amounts of sulfur species (Supplementary Fig. 1). The crystal structures of the typical catalysts were examined by X-ray diffraction (XRD) as shown in Supplementary Fig. 2. The main peaks are assigned to face-centered cubic (*fcc*) Ni (JCPDS: 04-0850), while other weak peaks are attributed to hexagonal close-packed (*hcp*) Ni emerging in low content[28]. Scanning electron microscopy (SEM) and transmission electron microscopy (TEM) images show that both catalysts with or without S modification are composed of highly dispersed Ni nanoparticles (NPs) with an average size of ~10 nm (Supplementary Figs. 3–5). High-resolution TEM (HRTEM) images show that the Ni NPs are well encapsulated, but not tightly, by the graphitized carbon layers with ~3 nm thickness (Supplementary Figs. 4–6), in which the interplanar distance of 0.203 nm is assigned to the (111) plane of fcc Ni. The Ni loadings were determined as 65.6–67.4 wt% for all Ni@C-$S_x$ catalysts by inductively coupled plasma–optical emission spectroscopy (ICP–OES) (Supplementary Table 1). These results suggest that there are no obvious differences in major parameters for the Ni@C-$S_x$ catalysts, such as crystalline phase, particle size, dispersion and Ni loading.

Elemental mapping through energy-dispersive X-ray spectrometry (EDS) shows that the C, S and Ni species are uniformly distributed throughout the surface of Ni@C-$S_{1/30}$ catalyst (Supplementary Fig. 4d–g). Notably, the S element is predominantly distributed around the Ni NPs, suggesting the strong coordination between Ni and S species. X-ray photoelectron spectroscopy (XPS) was then used to investigate the elemental compositions and chemical states on the catalyst surface (Supplementary Fig. 7). The XPS survey spectra clearly indicate that the S-doped Ni@C-$S_x$ catalysts contain Ni, C, S and O. The deconvoluted peaks of S 2$p$ spectra can be assigned to Ni–S (162.3 and 163.7 eV)[29], C–S species (164.1 and 165.1 eV)[30] and

oxidation of surface sulfur species (168.8 eV)[31]. In addition, the Ni 2$p_{3/2}$ spectra show that the dominant peaks at 853.0 eV can be assigned to metallic Ni[0], whereas the small peaks at 854.2 and 856.5 eV correspond to NiO and other Ni[2+] species (NiS$_x$, Ni(OH)$_2$ or NiCO$_3$), respectively[25,32–36]. It is noteworthy that the Ni/S ratios determined from XPS are substantially lower than those from ICP–OES and elemental analysis (Supplementary Table 1), implying that the sulfur species are mainly distributed on the catalyst surface. The sulfur-modified Ni surface is also proven by infrared spectroscopy with CO probe (Supplementary Fig. 8). Moreover, the XPS results of Ni@C-$S_{1/30}$ catalyst for different Ar$^+$ beam etching times further indicate that the sulfur species are concentrated at the interface of Ni surface and adjacent carbon layer (Supplementary Fig. 9). Taken together, these results confirm that sulfur has been successfully incorporated to the Ni surface.

### Catalytic performance

The catalytic performance of Ni@C-$S_x$ catalysts was evaluated in a stainless-steel autoclave reactor for the conversion of 50.0 wt% aqueous EtOH to LAS. Originally, the LAS selectivity is 49.2% (26.0% of $C_{8+}$OH) with an EtOH conversion of 51.8% over S-free Ni@C-$S_0$ catalyst at 180 °C (Fig. 2a and Supplementary Table 2). Meanwhile, the alcohol distribution follows the step-growth model, accompanied with a monotonic decrease in the alcohol selectivity with the increase of carbon number (Supplementary Fig. 10a). In order to explore the promotional effect of surface sulfur modification, a series of Ni@C-$S_x$ catalysts with different S/Ni ratios were screened systematically. As shown in Fig. 2a and Supplementary Table 2, a volcano-type correlation between EtOH coupling activity and S/Ni ratios was discovered, emerging with a summit at the S/Ni ratio of 1/30. Unexpectedly, we found that the appropriate incorporation of sulfur in the Ni@C catalyst not only prevents C−C bond cleavage and promotes EtOH conversion, but also regulates the alcohol distribution. Accordingly, the EtOH

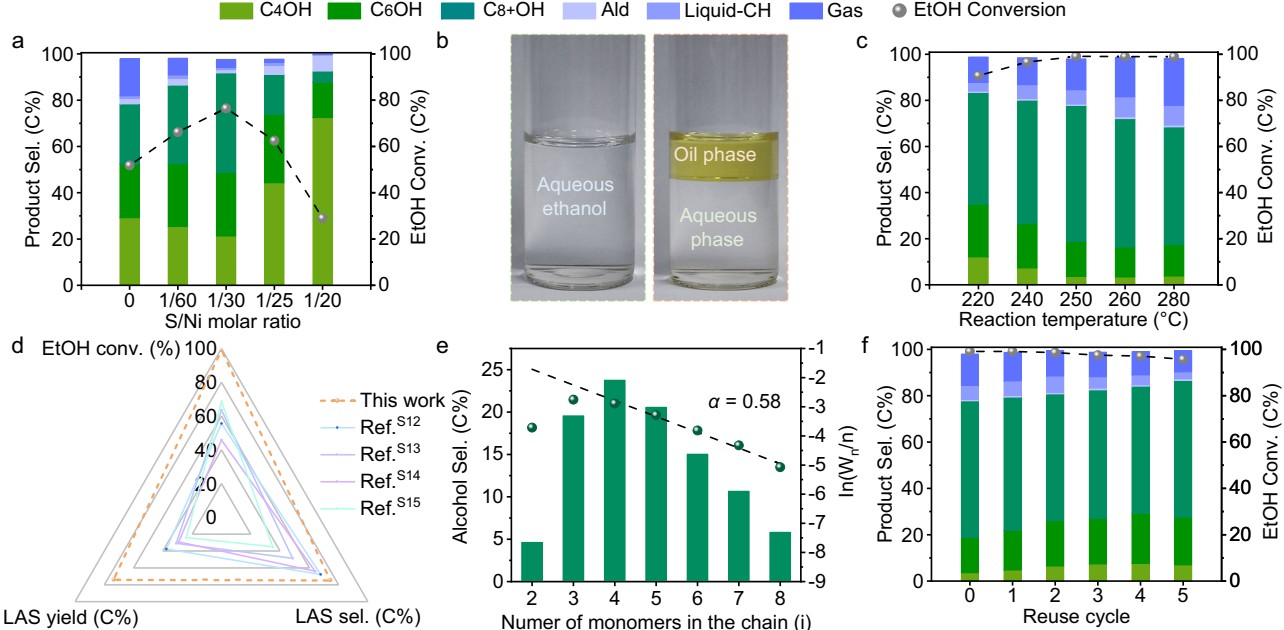

**Fig. 2 | Ethanol upgrading performance of Ni@C-$S_x$ catalysts. a** EtOH conversion and product selectivity over Ni@C-$S_x$ catalysts with different S/Ni molar ratios at 180 °C. **b** Photographs of aqueous EtOH feedstock (left) and obtained liquid products (right). **c** Catalytic performance of the Ni@C-$S_{1/30}$ catalyst in the two-stage intensification process. The reaction was performed at 180 °C and then at the second stage temperature for 6 h. **d** Catalytic performance comparison among Ni@C-$S_{1/30}$ catalyst and other previously reported catalysts. **e** Detailed alcohol distribution, step-growth plot and α value over Ni@C-$S_{1/30}$ catalyst under optimized reaction conditions. **f** Stability test for Ni@C-$S_{1/30}$ catalyst. Reaction conditions: 12 h, catalyst (0.3 g), NaOH (21.6 mmol), 50.0 wt% aqueous EtOH (108.0 mmol). (C$_4$OH: 1-butnaol, C$_6$OH: 1-hexanol and 2-ethyl-1-butanol, C$_{8+}$OH: C$_{8+}$ alcohols, Ald aldehydes, Liquid-CH liquid hydrocarbon products, Gas gaseous carbon products).

conversion and the LAS selectivity are remarkably enhanced up to 76.5% and 70.5% (43.0% of $C_{8+}OH$), respectively, over the optimal Ni@C-$S_{1/30}$ catalyst. In addition, the calculated chain growth probability ($\alpha$) value increases from 0.42 to 0.46 with the increase of sulfur content (Supplementary Fig. 10), which indicates the change of main products from *n*-butanol to LAS. Nevertheless, an excessively higher S/Ni ratio leads to a sharp decrease of EtOH conversion as well as LAS selectivity, as a result of blocking available Ni sites. As shown in Fig. 2b, oil−water stratification occurs naturally after reaction due to the strong hydrophobicity of LAS products. Among the two phases, the upper oil phase consists mainly of *n*-butanol and LAS (e.g., 2-ethyl-1-butanol, 1-hexanol and 2-ethyl-1-hexanol), together with very small quantities of alkanes and aldehydes, whereas the unreacted EtOH and water are primarily detected in the aqueous phase (Supplementary Figs. 11 and 12).

To further improve the performance, the reaction conditions were optimized over the Ni@C-$S_{1/30}$ catalyst. The effect of reaction time was first studied, similarly to the previous reports[16, 18,26,37,38]. The EtOH conversion gradually increases from 37.6 to 76.5% with the reaction time increasing from 1.5 to 12 h and remains almost unchanged in the next 12 h (Supplementary Fig. 13a and Supplementary Table 3). Given the endothermic property of dehydrogenation[15], increasing the reaction temperature will facilitate the further conversion of short chain alcohols (EtOH and *n*-butanol) to LAS. However, our results show that the C−C bond cleavage of EtOH happens easily at higher temperature, thereby lowering EtOH utilization efficiency and LAS selectivity (Supplementary Fig. 13b and Supplementary Table 4). For this reason, we proposed a new two-stage intensification process for this reaction to intensify EtOH coupling, which enabled the smooth conversion of EtOH into *n*-butanol and LAS at low-temperature stage and then further upgraded the unreacted EtOH and generated alcohols at high-temperature stage. Encouragingly, the catalytic performance of Ni@C-$S_{1/30}$ catalyst can be greatly enhanced (Fig. 2c and Supplementary Table 5), giving a LAS selectivity of 74.2% (15.2% of $C_6OH$ and 59.0% of $C_{8+}OH$) at an EtOH conversion of 99.1%. Notably, as compared with the reported results (Fig. 2d and Supplementary Table 6), the as-prepared Ni@C-$S_{1/30}$ catalyst exhibits both the highest EtOH conversion and LAS selectivity, particularly for $C_{8+}OH$. The alcohol distribution deviates significantly from the current step-growth model, and the LAS proportion is up to 96.4% among all alcohol products (Fig. 2e). The corresponding gas chromatography (GC) profiles of the collected oil/aqueous phase products show that the LAS products are almost completely separated from the aqueous phase (Supplementary Fig. 14). Moreover, the stability of the Ni@C-$S_{1/30}$ catalyst was examined as shown in Fig. 2f and Supplementary Table 7. The EtOH conversion and LAS selectivity are well maintained in successive six cycles. More importantly, XRD and XPS analyses show that the structure and surface chemical states of recycled catalysts remain almost unchanged (Supplementary Figs. 15 and 16), demonstrating the excellent stability of Ni@C-$S_{1/30}$ catalyst.

## Mechanistic studies

To understand the role of sulfur in tuning catalytic performance, we further revealed the key contributing factors for achieving the high-yield production of LAS in this reaction. The EtOH dehydrogenation rate, 2-butenal hydrogenation rate and C−C bond formation rate over catalysts were measured to acquire the kinetic information (Supplementary Tables 8–10). As regarded as the RLS of Guerbet condensation, the corresponding catalysts reveal relatively low-equivalent EtOH dehydrogenation rate in Fig. 3a. By contrast, sulfur modification dramatically reduces the catalytic hydrogenation rate of 2-butenal. Taken together with the rapid-swift catalysis process of forming C−C bond (Supplementary Table 11), these results suggest that the aldehydes generated from dehydrogenation are inclined to undergo the aldol condensation immediately. As such, the origin of high LAS yield should

be ascribed to the increased probability of direct growth during aldol condensation.

Based on the above results and previous studies[18, 39], we propose that the strong adsorption of generated aldehydes on catalyst surface is a key factor for improving the direct-growth probability through accumulation of aldehyde intermediates, thus improving the LAS selectivity. Considering that the *n*-butanal is an important intermediate in growth routes (Fig. 1), the *n*-butanal-temperature programmed desorption/mass spectroscopy (*n*-butanal-TPD/MS) was recorded by monitoring the peaks at $m/z = 44$ and 72 for typical Ni@C-$S_x$ catalysts. The peaks at $m/z = 44$ and 72 are attributed to *n*-propane and *n*-butanal, which are the dissociation fragment and molecular ion of *n*-butanal, respectively. As shown in Fig. 3b, c, the first desorption peaks for all catalysts appear below 200 °C and the curves of both $m/z = 44$ and 72 are detected simultaneously, suggesting that the first desorption peak corresponds to weak adsorption of *n*-butanal. Then, the other peaks attributed to the dissociation of adsorbed *n*-butanal can only be observed at $m/z = 44$ on the catalyst. Notably, a large dissociation peak in the region of 220–600 °C is observed for S-free Ni@C-$S_0$ catalyst while most dissociation peaks are observed in the region of 380–600 °C for Ni@C-$S_{1/30}$ catalyst. Additionally, we determined the Ni active sites on the Ni@C-$S_x$ catalysts. As shown in Supplementary Table 12, the number of Ni active sites decreases with increasing S/Ni ratio. The Ni active sites on Ni@C-$S_0$, Ni@C-$S_{1/30}$ and Ni@C-$S_{1/25}$ catalysts are 26.9 µmol g$^{-1}$, 15.9 µmol g$^{-1}$ and 7.8 µmol g$^{-1}$, respectively. Combining these results with the CO-DRIFTS observations (Supplementary Fig. 8), we believe that introduction of low-content sulfur to occupy part of the threefold hollow sites on Ni can not only effectively reduce the C−C bond cleavage, thereby increasing the temperature required for aldehyde dissociation, but also retain abundant strong adsorption sites for aldehydes simultaneously. However, the inconspicuous dissociation peak (~500 °C) for Ni@C-$S_{1/25}$ catalyst (Fig. 3c) and the ethanol upgrading results (Fig. 2a) suggest the low LAS selectivity is associated with the deactivation of adsorption sites, which can be ascribed to excess sulfur covering the active sites for aldehydes strong adsorption. Moreover, this correlation was again confirmed on NiSn@C and Ni@NC catalysts, with few strong adsorption peaks observed, whose alcohols distribution succumbs to the step-growth model (Supplementary Fig. 17). To sum up, the key to breaking the step-growth model is to tremendously retain the strong adsorption sites while inhibiting the C−C cleavage side reaction on the catalysts.

Furthermore, in situ diffuse reflectance-infrared Fourier-transform spectroscopy (DRIFTS) was employed to examine *n*-butanal adsorption on the typical Ni@C-$S_x$ catalysts (Fig. 3d–f). The bands at 1747 cm$^{-1}$, 2715/2811 cm$^{-1}$ and 2896/2971 cm$^{-1}$ which are assigned to the $v$(C=O) mode, aldehydic $v$(C−H) mode and aliphatic $v$(C−H) mode of adsorbed *n*-butanal[40–42], respectively, are observed over all Ni@C-$S_x$ catalysts at 50 °C. For Ni@C-$S_0$ catalyst, the bands at 2937, 2881, 1710, 1687 and 1643 cm$^{-1}$ display obvious increment as the temperature increases from 140 to 220 °C and then decrease gradually along with the further increase of temperature. The most obvious bands at 1687 and 1641 cm$^{-1}$ are ascribed to the $v$(C=O) and $v$(C=C) of absorbed 2-butenal, and the new bands at 2937/2881 cm$^{-1}$ and 1377/1461 cm$^{-1}$ attributed to $v$(C−H) and $\delta$(C−H) corresponding to adsorbed 2-butenal can also be observed[43–46]. The observation indicates that *n*-butanal molecules cannot be adsorbed on the Ni surface in stable state during the high-temperature desorption process, consistent with the TPD/MS results above. Surface sulfur modification can effectively stabilize the adsorption of *n*-butanal on Ni surface, and as such, the peaks corresponding to 2-butenal were no obvious over Ni@C-$S_{1/30}$ catalyst during the desorption process. Although the bands at 2937 and 2881 cm$^{-1}$ can also be observed as the temperature increases to 180 °C, the band for gaseous 2-butenal molecule has a greater absorbance than that for adsorbed 2-butenal (Supplementary Fig. 18)[45,46]. This suggests that

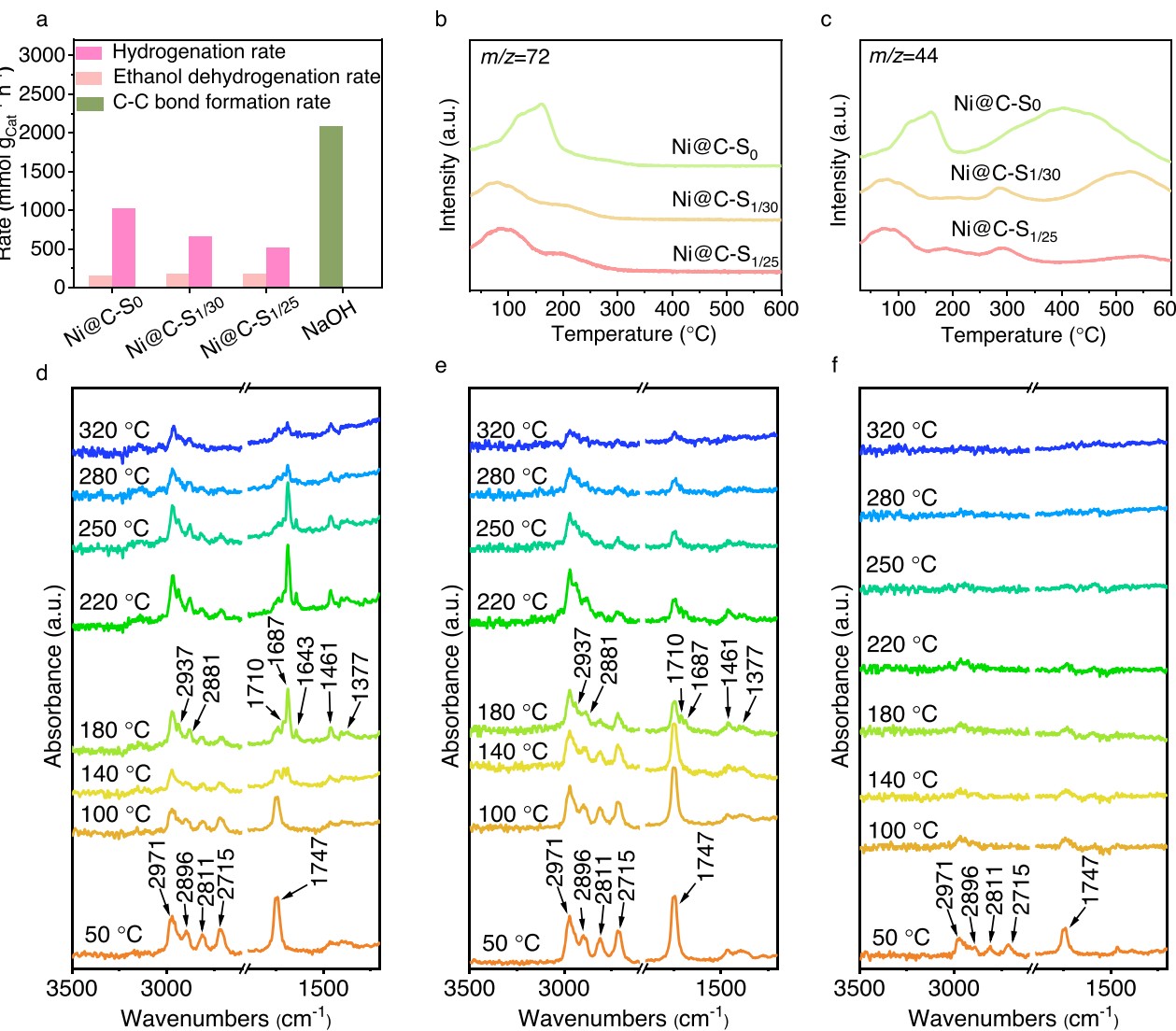

**Fig. 3 | Mechanism study of catalytic LAS production over Ni@C-$S_x$ catalysts.**
**a** The measured hydrogenation rate, EtOH dehydrogenation rate and C–C formation rate over catalysts. *n*-butanal-TPD/MS profiles over Ni@C-$S_x$ catalysts: Signal of *n*-butanal ($m/z = 72$) (**b**) and signal of 1-propane ($m/z = 44$) (**c**). Temperature-dependent in situ DRIFTS spectra for *n*-butanal adsorption on Ni@C-$S_0$ catalyst (**d**), Ni@C-$S_{1/30}$ catalyst (**e**) and Ni@C-$S_{1/25}$ catalyst (**f**).

most of the generated 2-butenal tends to desorb from Ni surface, thereby suppressing the further dissociation. Moreover, the adsorbed *n*-butanal can even be observed at 320 °C, confirming the existence of abundant strong adsorption sites over Ni@C-$S_{1/30}$ catalyst again. However, when excess sulfur was introduced to the catalyst, both dissociation and adsorption of aldehydes on Ni surface would be suppressed. As a result, the *n*-butanal was almost completely desorbed below 250 °C for Ni@C-$S_{1/25}$. Overall, the in situ DRIFTS results further demonstrate that the Ni@C-$S_{1/30}$ catalyst provides a stable strong adsorption Ni surface for aldehydes, thereby achieving the enrichment of aldehydes.

DFT calculations were also performed to explore the promoting origin of surface sulfur in the upgrading of aqueous ethanol to LAS. Firstly, optimized structures of Ni (111) surface with varying sulfur coverages were built to explore the adsorption behavior of *n*-butanal (Fig. 4a and Supplementary Figs. 19 and 20). DFT calculations indicate that the *n*-butanal molecule is adsorbed on the bridge site through the oxygen atom and the *n*-butanal molecule can still be stably adsorbed when there are several sulfur atoms around. The adsorption energies of *n*-butanal are in a narrow range of 1.21–1.34 eV as the sulfur coverage increases from 0 to 20%. However, the dense sulfur atoms covering the

Ni surface will hinder the adsorption of *n*-butanal. The adsorption energy sharply decreases from 1.26 to 0.87 eV when the sulfur coverage increases from 20 to 25% and then continues to decrease with the increase of sulfur coverage, in line with the results of surface sulfur coverage quantification (Supplementary Table 12), *n*-butanal-TPD/MS (Fig. 3b, c) and DRIFTS (Fig. 3d–f).

We then selected the Ni (111) surfaces covered with 0, 14 and 25% sulfur (denoted as Ni-S0%, Ni-S14% and Ni-S25%) as typical models to calculate the Gibbs free energies for the RLS of ethanol dehydrogenation and C–C bond cleavage, referring to previous research[13, 47–49]. The conversion of EtOH to acetaldehyde involves C–H and O–H bond breaking steps, among which the O–H bond breaking of $CH_3CH_2OH^*$ to $CH_3CH_2O^*$ is the RLS. Similarly to the adsorption behavior of *n*-butanal, the $CH_3CH_2OH^*$ and $CH_3CH_2O^*$ are also adsorbed on the Ni sites through the oxygen atom in the dehydrogenation process. For the dehydrogenation of EtOH, the dehydrogenation activation energy on Ni-S0%, Ni-S14% and Ni-S25% is 0.71, 0.70 and 0.80 eV, respectively. Interestingly, the dehydrogenation activation energy on Ni-S14% is similar to that on Ni-S0%, indicating that the dehydrogenation process is not sensitive to low sulfur coverage (Fig. 4b and Supplementary Fig. 21). However, for the RLS of C–C bond

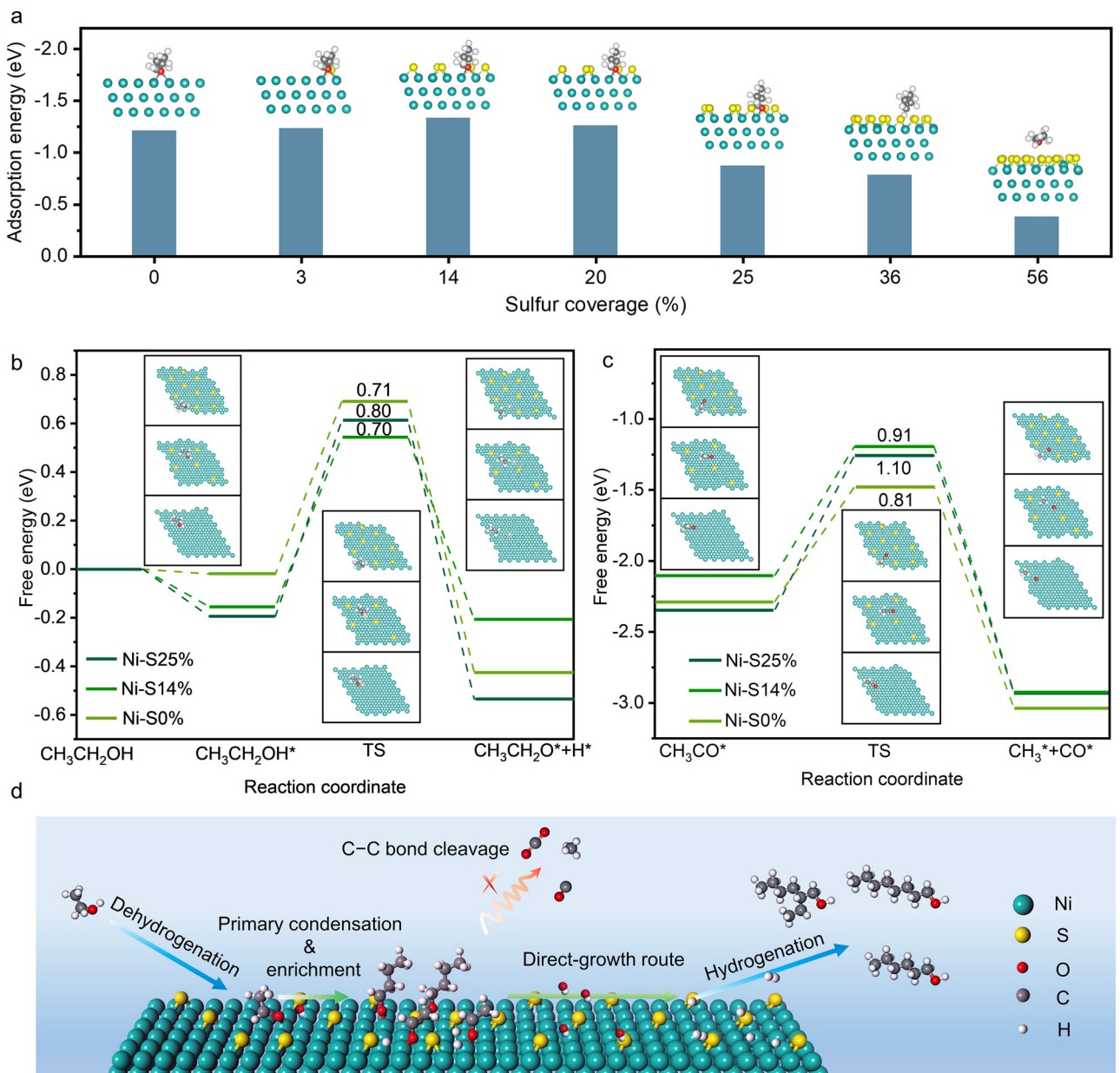

**Fig. 4 | DFT calculation results and reaction mechanism. a** Adsorption energy of *n*-butanal on the models of Ni (111) with different sulfur coverage. **b** Gibbs free energy diagrams for EtOH to $CH_3CH_2O^*$ on Ni (111) surfaces with different sulfur coverage. **c** Gibbs free energy diagrams for $CH_3CO^*$ to $CH_3^*$ and $CO^*$ on Ni (111) surfaces with different sulfur coverage. **d** Schematic illustration for the role of sulfur in promoting upgrading of EtOH to LAS.

cleavage, the activation barrier for $CH_3CO^*$ to $CH_3^*$ and $CO^*$ continues to increase from 0.81 to 1.10 eV with the increase of sulfur coverage from 0 to 25% (Fig. 4c and Supplementary Fig. 22), suggesting that the C–C bond cleavage on Ni surface with low sulfur coverage is effectively inhibited. Notably, the sulfur atoms occupying the threefold hollow sites on Ni (111) inhibit the adsorption of $CH_3^*$ and $CO^*$, which also need to occupy on the threefold hollow sites, thus hindering the C–C bond cleavage. The calculations are consistent with our CO adsorption (Supplementary Fig. 8) and experimental (Fig. 2a) results, demonstrating that small amount of sulfur occupying on threefold hollow sites can effectively inhibit the C–C bond cleavage but barely weaken the EtOH dehydrogenation.

Based on the results above, we propose a reaction scheme of EtOH upgrading to LAS over sulfur modified Ni surface as illustrated in Fig. 4d. This scheme indicates that EtOH molecules are dehydrogenated to acetaldehyde molecules and further upgraded to *n*-

butanal molecules. Meanwhile, the steric hindrance effect of sulfur on Ni surface inhibits the C–C bond cleavage of EtOH, reducing the consumption of NaOH and occupation of the active site, thereby promoting the aldolization of acetaldehyde molecules. In turn, the strongly adsorbed acetaldehyde and *n*-butanal molecules are enriched on Ni surface and finally converted to LAS through direct-growth route.

## Synthetic flexibility

Based on our proposed mechanism, the flexibility tests of surface sulfur modification strategy were also carried out. Firstly, we used another facile sulfur modification method to study the flexibility of adding method according to literature[50]. Dimethyl sulfoxide (DMSO) was directly introduced to the aqueous EtOH solution for in situ modifying Ni@C-$S_0$ catalyst during the reaction. The catalytic activity shows a volcanic-like variation tendency with the increase of DMSO

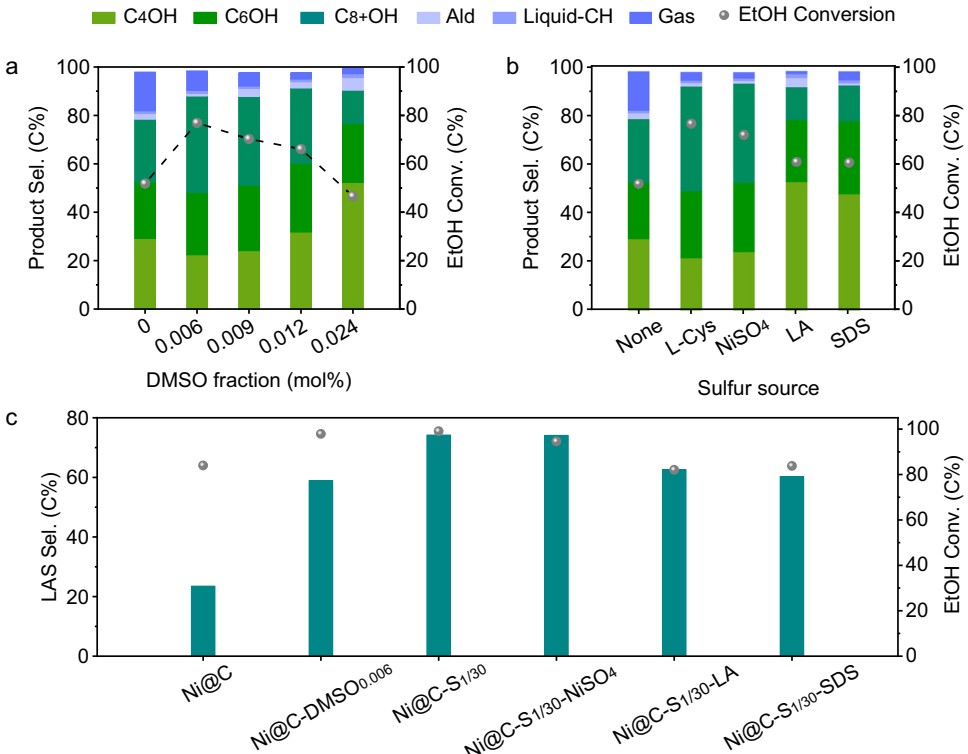

**Fig. 5 | Catalytic performance of catalysts prepared by different synthesis methods.** EtOH conversion and product selectivity over Ni@C-DMSO$_z$ catalysts with different DMSO fractions in aqueous EtOH (**a**) and Ni@C-S$_{1/30}$-y catalysts using different sulfur precursors (**b**). Reaction conditions: 180 °C, 12 h, catalyst (0.3 g), NaOH (21.6 mmol), 50.0 wt% aqueous EtOH (108.0 mmol). The catalytic performance of typical catalysts under the optimized reaction conditions (**c**). Reaction conditions: 180 °C for 6 h and then heating to 250 °C for 6 h, catalyst (0.3 g), NaOH (21.6 mmol), 50.0 wt% aqueous EtOH (108.0 mmol).

fraction, and the EtOH conversion and LAS selectivity reach maximum as the DMSO fraction increases to 0.006% (Fig. 5a, Supplementary Table 13 and Supplementary Fig. 23). Although XRD analysis shows that the catalyst after reaction has no difference with the original Ni@C-S$_0$ catalyst, the deconvoluted XPS peaks assigned to S–Ni and S element distributed around the Ni NPs can also be observed over the Ni@C-DMSO$_{0.006}$ (Supplementary Figs. 24–26), suggesting that the sulfur can be directly modified on the Ni surface.

Then, the flexibility of sulfur precursor was also evaluated. In addition to L-cysteine (L-Cys), NiSO$_4$, lipoic acid (LA) and sodium dodecyl sulfate (SDS) were also used as sulfur precursors with the same S/Ni ratio of 1/30. As revealed in Fig. 5b, Supplementary Table 14 and Supplementary Fig. 27, the catalyst using NiSO$_4$ as a precursor shows a high selectivity for LAS (28.5% for C$_6$OH and 40.7% for C$_{8+}$OH). Additionally, the catalysts using LA and SDS as precursors have higher poisoning degree, whose catalytic performances are similar to that of Ni@C-S$_{1/25}$ catalyst. Those results show that the change of the sulfur precursors can also effectively enhance the EtOH conversion and alcohol selectivity, but there are some differences in poisoning degree between various sulfur precursors.

Under the optimized reaction temperature of 250 °C, the selectivity of LAS over S-free Ni@C-S$_0$ catalyst is only 23.5%. The catalyst performance of Ni@C-S$_0$ catalyst is even lower than that at low temperature (180 °C) due to the more severe C–C bond cleavage. In contrast to Ni@C-S$_0$ catalyst, both Ni@C-S$_{1/30}$-y (y stands for sulfur precursor) and optimal Ni@C-DMSO$_{0.006}$ catalysts exhibit desirable LAS selectivity (>58.0%) at high conversions of EtOH (>80.0%) (Fig. 5c and Supplementary Table 15). Moreover, compared to Ni@C-S$_0$ catalyst, the deviation of the alcohol distribution from the step-growth model is more obvious over the Ni@C-S$_{1/30}$-y and optimal Ni@C-DMSO$_{0.006}$ catalysts (Supplementary Fig. 28), implying that the inhibition of the C–C bond cleavage is a prerequisite for improving the

selectivity of LAS. In brief, the above results confirm the flexibility of surface sulfur modification for preparing Ni@C-S$_x$ catalysts in terms of adding method and sulfur precursors.

## Discussion

In summary, we have demonstrated that the controllable exposure of Ni sites by surface sulfur modification in the case of Ni@C-S$_x$ catalysts can greatly accelerate the aqueous EtOH upgrading to LAS. The optimized Ni@C-S$_{1/30}$ catalyst exhibited a high LAS selectivity of 74.2% (15.2% of C$_6$OH and 59.0% of C$_{8+}$OH) with a 99.1% of EtOH conversion. To our knowledge, this is an impressive breakthrough for realizing EtOH upgrading to LAS, achieving the record-high EtOH conversion and LAS selectivity to date. The in situ characterization and DFT calculation results reveal that the low content of sulfur selectively shelters the active Ni sites from C–C bond cleavage via steric hindrance, and retains abundant strong adsorption sites for aldehyde intermediates simultaneously, thereby improving the direct-growth probability to boost LAS production. In addition, the flexibility of surface sulfur modification was also demonstrated by changing modifying approaches and sulfur precursors. This work offers an innovative paradigm for the rational design of high-performance catalysts toward aqueous EtOH upgrading by effective regulation of surface active sites.

## Methods

### Synthesis of Ni@C-S$_x$ catalysts

Ni(NO$_3$)$_2$·6H$_2$O (23.7 mmol), CAM (47.5 mmol) and an appropriate amount of L-Cys (0/0.395/0.79/0.948/1.185 mmol) were dissolved in 50.0 ml of deionized water and stirring until completely dissolved. The mixture was gelled gradually by evaporation at 80 °C and then dried at 100 °C until fully dried. Then the dried gels were carbonized at 550 °C for 2 h with a 5 °C min$^{-1}$ heating rate under N$_2$ atmosphere to obtain the

catalysts. The as-prepared catalysts were denoted as Ni@C-S$_x$, where x represented the molar ratio of sulfur (S) to nickel (Ni).

### Synthesis of Ni@C-DMSO$_z$ catalysts

DMSO was directly introduced to the aqueous EtOH solution for in situ modifying Ni@C-S$_0$ catalyst during the reaction. After the reaction, the used catalyst was fully washed by EtOH and then dried in vacuum for 12 h. The catalyst was denoted as Ni@C-DMSO$_z$, where z represented the mole fraction of DMSO added to the EtOH aqueous solution.

### Synthesis of Ni@C-S$_{1/30}$-y catalysts

Ni@C-S$_{1/30}$-y catalysts were synthesized by following the typical protocol for the synthesis of Ni@C-S$_x$ catalysts with a S/Ni molar ratio of 1/30, except that the sulfur precursor of L-Cys was replaced by the same molar amount of NiSO$_4$, LA or SDS, respectively. The y represented the abbreviation name of sulfur precursor.

### Characterizations

XRD patterns in the range of 2 theta = 10–80° were recorded on a Rigaku Mini Flex 600 X-ray diffractometer with Cu-K$\alpha$ ($\lambda$ = 0.15406 nm). Brunauer-Emmett-Teller (BET) experiments were tested on a MicroActive TriStar II 3020 analyzer at −196 °C. The Ni contents in the catalysts were determined by ICP−OES with a PerkinElmer Optima 8000 instrument. Sulfur and carbon contents in the catalysts were determined on a Vario El cube elemental analyzer. The morphologies of samples were performed by SEM (Hitachi SU8010) and TEM (Talos F200S). EDS mappings were carried out on FEI Talos F200X, equipped with Super X-EDS system (four systematically arranged windowless silicon drift detectors). XPS measurements were performed on a Thermo Scientific Escalab 250XI, and all the binding energies were calibrated by the C 1$s$ peak at 284.8 eV. The depth profile was obtained by etching the Ni@C-S$_{1/30}$ catalyst with Ar$^+$ ion gun at 500 eV.

### Aqueous EtOH upgrading experiments

Upgrading of aqueous EtOH was carried out in a 70 ml stainless-steel autoclave reactor (SSC70*4-Y2W6-316-GY, Jiangsu Safety Instrument Co., Ltd, Nanjing, China). Typically, the catalyst (0.3 g) and NaOH (21.6 mmol) were added to 50.0 wt% aqueous EtOH (108.0 mmol) in the reactor with a magnetic stirrer bar. The inner atmosphere was then repeatedly evacuated and recharged with H$_2$ (0.1 MPa) five times. The reactor was subsequently heated to the specified temperature (140–260 °C) for the required time (1.5–24 h). The stirring rate was 1000–1200 rpm. Note: Reactions using the two-stage intensification approach require changing the heating program to hold the reaction at 180 °C for 6 h and then heat the system to the second temperature (220–280 °C) for another 6 h. After the reaction, the reactor was immediately cooled with a water bath to room temperature. To test the catalyst stability, the spent catalyst was separated by centrifugation and washed with deionized water and acetone three times, respectively. Then the catalyst was dried in vacuum at room temperature for 12 h and then directly used for the next run.

### In situ DRIFTS for CO adsorption

The adsorption of CO on the catalysts was monitored by in situ DRIFTS (Bruker IFS 66 v) equipped with Harrick diffuse reflectance accessory with ZnSe and quartz window at BL01B in the NSRL in Hefei, China. Each spectrum was recorded by averaging 128 scans at a resolution of 2 cm$^{-1}$. Before the measurements, the sample wafer was pretreated under a flow of 10% H$_2$/Ar for 1 h at 450 °C to remove the impurities absorbed on the surface, which was then switched to Ar flow and cooled to room temperature. The background was collected at room temperature under the flow of Ar (30 ml min$^{-1}$) before the adsorbate introduction. Subsequently, CO was introduced into the system for

10 min and then swept under pure Ar for 10 min to remove the gaseous CO. The spectra were collected by averaging 128 scans at a resolution of 2 cm$^{-1}$.

### In situ DRIFTS for *n*-butanal adsorption

The adsorption of *n*-butanal on the catalysts was monitored by DRIFTS (Bruker IFS 66 v) equipped with Harrick diffuse reflectance accessory with ZnSe and quartz window at BL01B in the NSRL in Hefei, China. Each spectrum was recorded by averaging 128 scans at a resolution of 2 cm$^{-1}$. Before the measurements, the sample wafer was pretreated under a flow of 10% H$_2$/Ar for 1 h at 450 °C to remove the impurities absorbed on the surface, which was then switched to Ar flow and cooled to 50 °C. During the cooling process, the background spectra at temperatures of 320, 280, 250, 220, 180, 140, 100 and 50 °C were recorded, respectively. Subsequently, *n*-butanal absorbed on the catalyst surface until saturation by introducing the *n*-butanal vapor into infrared cell. Then the system was purged with a flow of Ar (30 30 ml min$^{-1}$). Finally, DRIFTS spectra were recorded at 50 °C after stepwise heating to 100, 140, 180, 220, 250, 280 and 320 °C.

### *n*-Butanal-TPD/MS experiments

The n-butanal-temperature programmed desorption/mass spectroscopy (*n*-butanal-TPD/MS) measurements were carried out using a micro-reactor with a Pfeiffer OmniStar ThermoStar mass spectrometer. The sample of 0.1 g was reduced at 450 °C for 1 h in 10% H$_2$/Ar flow in a quartz reactor, which was then switched to Ar flow and cooled to room temperature. Then *n*-butanal absorbed on the catalyst for 3 h by introducing the *n*-butanal vapor into quartz reactor. The sample was purged by Ar flow (30 ml min$^{-1}$) at room temperature for 1 h, and then was heated from room temperature to 600 °C (10 °C min$^{-1}$). The signal of *n*-butanal (m/z = 72) and 1-propane (m/z = 44) were monitored with MS.

### Computational details

All the spin-polarization DFT calculations were conducted based on the Vienna ab initio simulation package (VASP)[51,52]. The electron-ion interactions were described by the Projected Augmented-Wave (PAW) potentials, while the exchange-correlation interactions were calculated by employing the Perdew−Burke−Ernzerhof (PBE) functional of Generalized Gradient Approximation (GGA)[53,54]. The vdW-D3 method developed by Grimme was employed to describe the van der Waals interaction[55]. The plane-wave energy cutoff was set as 400 eV. The convergence threshold was set as $1.0 \times 10^{-4}$ eV in energy and 0.02 eV per Angstrom in force. A vacuum layer of at least 15 Å was adopted to avoid the periodic interactions. The Brillouin zone was modeled by gamma centered Monkhorst−Pack scheme, in which $7 \times 7 \times 7$ and $2 \times 2 \times 1$ grids were adopted for bulk and all slab models. The models were constructed as following: first of all, Ni(FM3M) bulk geometry optimization was performed. Then Ni (111) slab model consisting of three layers ($3 \times 3$ supercell and 108 atoms in total) was constructed based on the optimized bulk configuration. As for the S-supported slab model, numbers of S atoms were added on the Ni (111) slab model. The adsorption energies of *n*-butanal were calculated based on the following formula:

$$E_{ads}(BA) = E_{BA^*} - E^* - E_{BA} \qquad (1)$$

where the asterisk denotes the S supported slab model, *n*-butanal is abbreviated as BA, and $E_{BA^*}$, $E^*$ and $E_{BA}$ represent the total energies of slab models with one adsorbed BA molecule, the slab models alone and one BA molecule, respectively. The transition states of EtOH dehydrogenation and C−C bond cleavage were found by combination of CINEB and DIMER method. The change in the Gibbs free energy

($\triangle G$) of each reaction was calculated using the following equation:

$$\Delta G = \Delta E_{\text{pot}} + \Delta E_{\text{ZPE}} - T\Delta S \qquad (2)$$

in which the $\Delta E_{\text{pot}}$, $\Delta E_{\text{ZPE}}$ and the $\Delta S$ were referred to as the change in potential energy, the change in the zero-point energy and the change in the entropy, respectively. The zero-point energy was calculated by the summation of all vibrational frequencies:

$E_{\text{ZPE}} = \frac{1}{2}\sum h\upsilon$, where the $\upsilon$ corresponds to the vibrational frequency of each normal mode[56].

## Reporting summary
Further information on research design is available in the Nature Portfolio Reporting Summary linked to this article.

## Data availability
The data that support the findings of this study are available within the paper and its Supplementary Information, and all data are also available from the corresponding authors upon reasonable request. Source data are provided with this paper.

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

## Acknowledgements
This work was supported by the National Nature Science Foundation of China (U21A20288, 22278088, 22109148, 51902311, 21725102), the National Key Research and Development Project (2018YFE0125200), and the Guangdong Provincial Key Laboratory of Plant Resources Biorefinery (2021GDKLPRB-K04).

## Author contributions
S.Q., R.L., T.W. and Y.X. supervised the project. J.G., S.Q., W.G., Q.Z., T.W. and Y.X. conceived and designed the experiments. J.G. and Q.Z. performed the key experiments and analyzed the results. J.M., X.W., Jiawei L., Jiayi L., Y.F. and X.Z. assisted to carry out the in situ DRIFTS and TPD/MS characterization. J.G., W.G., Q.Z., S.Q., T.W. and Y.X. co-wrote the manuscript. All the authors discussed the results and commented on the manuscript.

## Competing interests
A patent application was submitted to China National Intellectual Property Administration (patent applicant: Guangdong University of Technology; names of inventors: T.W., J.G., SQ, Q.Z., Xiaoping Wu; application number: 202111585174.5; status of application: submitted; specific aspect of manuscript covered in the patent application: the preparation of Ni@C-S$_x$ catalysts). Other than this, the authors declare no competing interests.
