## [Peer Review File · Nature Communications]

REVIEWER COMMENTS

Reviewer #1 (Remarks to the Author):

Comments on Manuscript of NCOMMS-23-30660-T

Upgrading ethanol to long-chain alcohols provides a sustainable approach with carbon neutrality from biomass, as well as a flexible approach from fossil resources (syngas). It can produce high-quality biofuels and environment/human friendly oxygenated fine chemicals. The topic is important. Authors prepared the sulfur modified Ni catalysts and evaluated the performance. The optimal catalyst achieved both high ethanol conversion of 99.1% and high long-chain alcohols selectivity (15.2% C6 alcohols and 59.0% C8+ alcohols). The performance is high. The Anderson-Schulz-Flory (ASF) distribution often occurs in Fischer-Tropsch-like type reactions. It is of great significance to break through this ASF distribution to achieve a desired product distribution. Authors also investigated structure-effect relationship by both in-situ characterizations and theoretical simulation. This research will provide helpful guidance on understanding the long-chain alcohols growth. It seems to the reviewer that the manuscript is informative and well organized. The data are presented in detail. The English writing is also quite good, and the readability is good. I would like to recommend it to be accepted after some necessary revisions. The following suggestions may help authors to further improve the quality of this manuscript.

(1) Prefix, suffix and variables should be italicized. E.g, Line 49, n-butanol.

(2) Line 60, in the literature, although the optimal catalytic performance was achieved at Ni loading of 0.25 wt.% (LAS selectivity up to 67.7% with 55.6% of EtOH conversion), the Ni loading can reach 1 wt.% while still showing product distribution substantially higher than the Anderson-Schulz-Flory distribution.

(3) Line 99, the sentence of "...which allows various sulfur precursor selections and adding ways." may be rewritten as "...which allows various sulfur precursors selection and adding methods".

(4) Lines 113-114, the author mentioned that "the main peaks are assigned to face-centered cubic (fcc) Ni (JCPDS: 04-0850), while other weak peaks are attributed to hexagonal close-packed (hcp) Ni emerging in low content". Which form of Ni is the key active sites? The fcc Ni or the hcp Ni? Why?

(5) Lines 118-119, the author mentioned that "the Ni NPs are well encapsulated in the graphitized carbon layers with ~3 nm thickness". It may be better to indicate how the reactants interact with the Ni centers. Some readers may have questions because Ni NPs are well encapsulated in the graphitized carbon layers with ~3 nm thickness.

(6) Lines 121-124, the authors mentioned that "... were determined as 65.6–67.4 wt% for all Ni@C-Sx catalysts by inductively coupled plasma–optical emission spectroscopy (ICP–OES) (Supplementary Table 1). These results suggest that there are no obvious differences in major parameters for the Ni@C-Sx catalysts." . And the Supplementary Table 1 shows that the Ni/S ratio of the sample of Ni@C-S1/30 was 28.4 at.% by ICP OES analysis and 5.7 at.% by XPS analysis. On the one hand, the difference may be caused by the enrichment of S on the surface of the catalyst. But both of these two numbers are significantly greater than the ratio of 1/30 when preparing the catalyst.

(7) In this manuscript, the S content affects the catalytic performance, so it is very important to determine this Ni/S ratios, especially the Ni/S ratio of the surface (5.7 at.% by XPS analysis), and they may affect the validity of DFT computational model. Please check.

(8) Lines 132-133, the authors mentioned that "The deconvoluted peaks of S 2p spectra can be assigned to Ni–S (162.3 and 163.7 eV)". Is the Ni–S Sulfide or Sulfate?

(9) Lines 180-183, the authors mentioned that "Among the two phases, the upper oil phase consists mainly of 1-butanol and LAS (e.g., 2-ethyl-1-butanol, 1-hexanol and 2-ethyl-1-hexanol), together with very small quantities of alkanes and aldehydes, whereas the unreacted EtOH and water are primarily detected in the aqueous phase". The catalysts stay in which phase in the reaction?

(10) Line 229, please explain the rapid-swift rate in the sentence of "...Taken together with the rapid-swift rate in C–C bond formation...".

(11) Line 308, why the Ni (111) surfaces covered with 0%, 14% and 25% sulfur was selected for the DFT computational model?

(12) Lines 358-360 and Supplementary Fig. 24 , the authors mentioned that "...the deconvoluted XPS peaks assigned to S–Ni and S element distributed around the Ni NPs can also be observed over the Ni@C-DMSO.006 ...". Are the S species in this case Sulfides or Sulfates? Their status should be clarified to justify the explanations in the discussion.

(13) Lines 405-406, the authors mentioned that "...the dried gels were carbonized at 550 oC for 2 h with a 5 oCmin⁻¹ heating rate under N₂ atmosphere to obtain the catalysts." in the synthesis of Ni@C-Sx catalysts. But "... the fully dried solid gels were annealed at 600 oC for 4 h with a heating rate of 5 oC min⁻¹ under N₂ atmosphere" in the synthesis of NiSn@C catalyst. And "... the dried gel was carbonized at 500 oC for 2 h with a heating rate of 10 oC min⁻¹ under N₂ atmosphere." in the synthesis of Ni@NC catalyst. Is there anything special about the choices of heat treatment parameters here to achieve a certain goal?

(14) Line 521, The citation format needs to be improved.

(15) Is it Ni@C-S1/30 in the XPS spectra for Ni@C-Sx catalysts Supplementary Fig. 15 ?

(16) Please clarify the reaction conditions of "under optimized conditions" for Supplementary Fig. 27 in SI.

(17)Page 41, under the Supplementary Table 6, it may be "Conversion ×Selectivity".

(18)Page 45, under the Supplementary Table 10, it may be “Calculated from the results in Supplement Table S 8 and Table S9”.

(19) Page 45, under the Supplementary Table 10, why the hydrogenation rate was much higher than the dehydrogenation rate ?

Reviewer #2 (Remarks to the Author):

In this manuscript, the authors developed a catalyst with controllably exposed Ni catalytic sites on the surface through sulfur modification. This modification effectively stabilizes and enriches aldehyde intermediates, leading to a significant improvement in the direct-growth probability of LAS production. While the study includes routine work and theoretical insights from DFT calculations about the varying sulfur coverages' role in 1-butanol adsorption, the experimental demonstration of the structure-activity relationship remains unclear. Furthermore, the authors' previous work (Applied Catalysis B: Environmental, 2023, 321, 122048) has already explored the modification of aldehyde intermediate adsorption for C-C bond cleavage over Ni catalysts, which may reduce the novelty of the current study. The detailed comments are shown below :

(1) The authors have proposed two routes for ethanol coupling, namely a step-growth route and a direct-growth route. In these pathways, 1-butanaldehyde serves as the key intermediate and undergoes either hydrogenation or condensation reactions depending on the concentration of the intermediate aldehyde. Therefore, I believe it is not appropriate to attribute the high yield of long-chain alcohols to differences in routes. Additionally, butanol is typically regarded as a fully hydrogenated molecule that presents greater difficulty in dehydrogenation, so the formation of long-chain alcohols is still mainly through the condensation of intermediate aldehydes. The authors should consider this point carefully.

(2) How does Ni maintain a high dispersion at such a high loading (~65%) in Ni@C-Sx catalysts. And is this affected by the support or sulfur modification?

(3) In Fig. 2a, the catalysts with different S/N molar ratios exhibited diverse reaction activities, with the Ni@C-S1/30 catalyst having the best ethanol conversion and long chain alcohol selectivity. It can be noted that the Ni@C-S1/25 catalyst has higher conversion and higher butanol selectivity than the sulfur-free catalyst. However, in the in situ DRIFTS spectra of Figure 3d-f, the Ni@C-S1/25 catalyst exhibited a completely different spectral signals. In Figure 3f, we even do not observe the obvious signal of butanol over 50 °C. More explanations are required.

(4) The author's algorithm for C-C bond formation rate may present a potential issue. In my opinion, it would be counted in terms of the product, not the ethanol of the reaction, as not all products are exclusively coupling products.

(5) The authors use 50 wt% aqueous ethanol solution as a feedstock, why not feed higher purity ethanol in this case? Additionally, would water potentially have any role in the catalytic reaction or catalyst?

(6) From the DFT calculation, the adsorption energy of butanal on sulfur-free Ni@C and sulfur-rich Ni@C catalysts are pretty close. How to explain the large ethanol conversion difference between Ni@C-S1/30 catalyst (76.5 %) and sulfur-free Ni@C (only 51.8 %) at 180 °C.

(7) It is noted that the carbon balance of the different catalysts is below 90%. How about carbon deposition of the catalyst, please give the TG profiles after the reaction.

(8) The authors mentioned, "the introduction of low-content sulfur can not only effectively reduce the number of active sites for C–C bond cleavage and thus increase the temperature required for aldehyde dissociation, but also retain abundant adsorption sites for aldehydes simultaneously". If possible, it is recommended that the number of active sites is quantified to facilitate a better structure-activity relationship.

Reviewer #1:

Upgrading ethanol to long-chain alcohols provide a sustainable approach with carbon neutrality from biomass, as well as a flexible approach from fossil resources (syngas). It can produce high-quality biofuels and environment/human friendly oxygenated fine chemicals. The topic is important. Authors prepared the sulfur modified Ni catalysts and evaluated the performance. The optimal catalyst achieved both high ethanol conversion of 99.1% and high long-chain alcohols selectivity (15.2% C₆ alcohols and 59.0% C₈₊ alcohols). The performance is high. The Anderson-Schulz-Flory (ASF) distribution often occurs in Fischer-Tropsch-like type reactions. It is of great significance to break through this ASF distribution to achieve a desired product distribution. Authors also investigated structure-effect relationship by both in-situ characterizations and theoretical simulation. This research will provide helpful guidance on understanding the long-chain alcohols growth. It seems to the reviewer that the manuscript is informative and well organized. The data are presented in detail. The English writing is also quite good, and the readability is good. I would like to recommend it to be accepted after some necessary revisions.

We really appreciate the referee's highly positive evaluation for our work, and are very grateful to the referee for his/her insightful suggestions to help us further improve the quality of our manuscript.

The following suggestions may help authors to further improve the quality of this manuscript.

1. Prefix, suffix and variables should be italicized. E.g, Line 49, n-butanol.

We thank the referee for bringing this to our attention. In the revision, we have now modified the prefix, suffix and variables in the revised manuscript and SI.

2. Line 60, in the literature, although the optimal catalytic performance was achieved at Ni loading of 0.25 wt.% (LAS selectivity up to 67.7% with 55.6% of EtOH conversion), the Ni loading can reach 1 wt.% while still showing product distribution substantially higher than the Anderson-Schulz-Flory distribution.

We thank and agree with the referee for his/her insightful comment. According to the data in Figure 9 of the literature (ACS Sustain. Chem. Eng. **10**, 3466-3476 (2022)), the selectivity of C₆₊ alcohols is marginally lower at 1 wt% Ni loading compared to 0.25 wt% Ni loading (~60% vs. ~65%). However, the product distribution of the Ni/bio-apatite

catalyst with 1 wt% Ni loading is still greater than that predicted by the Anderson-Schulz-Flory distribution. Therefore, we have modified the statement from “...the Ni loading was limited to 0.25 wt%...” to “...the Ni loading was limited within 1 wt%...” in the revised manuscript.

3. Line 99, the sentence of “...which allows various sulfur precursor selections and adding ways.” may be rewritten as “...which allows various sulfur precursors selection and adding methods”.

We thank the referee for his/her valuable suggestion. According to the suggestion, we have now modified “which allows various sulfur precursor selections and adding ways” to “which allows for flexibility in the selection of various sulfur precursors and adding methods” in the revised manuscript.

4. Lines 113-114, the author mentioned that “the main peaks are assigned to face-centered cubic (fcc) Ni (JCPDS: 04-0850), while other weak peaks are attributed to hexagonal close-packed (hcp) Ni emerging in low content”. Which form of Ni is the key active sites? The fcc Ni or the hcp Ni? Why?

We thank the referee for his/her thoughtful question. Based on our previous studies (*Chem. Eng. J.* **453**, 139583 (2023)); *Appl. Catal. B.* **321**, 122048 (2023)), it can be objectively stated that the crucial active sites in the catalysts originate from the fcc Ni. Moreover, as shown in Fig. R1, we have prepared the Ni@C-S_{1/30} catalyst with a pure fcc Ni phase, named Ni@C-S_{1/30}-pure fcc, to validate our hypothesis. As described in the previous study (*RSC Adv.* **4**, 19488 (2014)), the hcp phase of Ni is an unstable phase that can be transformed into the stable fcc Ni phase by extending the carbonization time. Hence, we obtained the Ni@C-S_{1/30}-pure fcc catalyst by increasing the carbonization time from 2 h to 4 h, and evaluated its catalytic ability to upgrade ethanol. As shown in Fig. R2 and Table R1, although the ethanol conversion of the Ni@C-S_{1/30}-pure fcc catalyst (71.3%) is slightly lower than that of the Ni@C-S_{1/30} catalyst (76.1%), the selectivity of LAS remains high at 77.5%. This clearly indicates that the active Ni sites in the as-prepared Ni@C-S_{1/30} catalyst are predominantly derived from the fcc Ni.

Yet it is noteworthy that the hcp Ni phase could play a positive role in promoting ethanol conversion. Hence, it is worthwhile to carry out an in-depth study. We aim to fabricate a catalyst consisting of the pure hcp Ni phase and systematically document the promotion mechanism in our forthcoming research.

Fig. R1 | XRD patterns of Ni@C-S_{1/30} and Ni@C-S_{1/30}-pure fcc catalysts.

Fig. R2 | Comparison of ethanol coupling over the Ni@C-S_{1/30} and Ni@C-S_{1/30}-pure fcc catalysts.

Table R1 | Comparison of ethanol coupling over the Ni@C-S_{1/30} and Ni@C-S_{1/30}-pure fcc catalysts.

Catalyst	Conversion (%)	Carbon balance (%)	Selectivity (C%)						
			C ₄ OH	C ₆ OH	C ₈₊ OH	Ald	L-CH	Gas	Other
Ni@C-S _{1/30}	76.1	85.9	21.6	26.6	46.2	0.7	0.9	3.2	0.8
Ni@C-S _{1/30} -pure fcc	71.3	84.5	17.6	24.4	53.1	1.3	0.8	1.0	1.7

^aReaction conditions: 180 °C, 12 h, Ni@C-S_{1/30} catalyst (0.3 g), NaOH (21.6 mmol), 50.0 wt% aqueous EtOH (108.0 mmol).

5. Lines 118-119, the author mentioned that “the Ni NPs are well encapsulated in the graphitized carbon layers with ~3 nm thickness”. It may be better to indicate how the reactants interact with the Ni centers. Some readers may have questions because Ni NPs are well encapsulated in the graphitized carbon layers with ~3 nm thickness.

We thank the referee for his/her insightful suggestion. In fact, the Ni NPs are not entirely enclosed in the graphitized carbon layers, meaning that some active Ni sites on the Ni NPs are exposed to reactants. This allows for the adsorption and reaction of ethanol molecules. To visually showcase the abundance of exposed active Ni sites on the catalyst, the Ni@C-S_{1/30} catalyst was introduced to an excess hydrochloric acid solution (1.0 M HCl) and kept standing for 30 min. According to the TEM images of Ni@C-S_{1/30} catalyst after acid treatment (Fig. R3), only a small number of Ni NPs was tightly wrapped by the graphitized carbon layers, whereas the majority of the exposed Ni NPs were etched and completely eliminated by 1.0 M HCl. As a result, numerous honeycomb-like holes were remained on the carbon layers.

To enhance clarity and objectivity, we have revised the original statement “the Ni NPs are well encapsulated in the graphitized carbon layers with ~3 nm thickness” to “the Ni NPs are well encapsulated, but not tightly, by the graphitized carbon layers with ~3 nm thickness”. Furthermore, we have supplemented additional TEM images of Ni@C-S_{1/30} catalyst after acid treatment in Supplementary Fig. 6.

Fig. R3 | TEM images of the Ni@C-S_{1/30} catalyst after treated by 1.0 M HCl.

6. Lines 121-124, the authors mentioned that “... were determined as 65.6–67.4 wt% for all Ni@C-S_x catalysts by inductively coupled plasma–optical emission spectroscopy (ICP–OES) (Supplementary Table 1). These results suggest that there are no obvious differences in major parameters for the Ni@C-S_x catalysts.”. And the Supplementary Table 1 shows that the Ni/S ratio of the sample of Ni@C-S_{1/30} was 28.4 at.% by ICP-OES analysis and 5.7 at.% by XPS analysis. On the one hand, the difference may be caused by the

enrichment of S on the surface of the catalyst. But both of these two numbers are significantly greater than the ratio of 1/30 when preparing the catalyst.

We thank the referee for his/her thoughtful comment. We apologize for any confusion caused by our mistake in the Supplementary Information. In Supplementary Table 1, the Ni/S ratio should be given as an atomic ratio instead of atomic percentage. In the case of the Ni@C-S_{1/30} sample, 28.4 of the Ni/S ratio was calculated according to the experimental results from both elemental analysis and ICP-OES analysis, while 5.7 of the Ni/S ratio was calculated using XPS analysis. To avoid any further misunderstanding, we have now corrected this mistake in Supplementary Table 1.

As the referee pointed out in the comment, it should be noted that the Ni/S ratio of 5.7 is significantly lower in the XPS analysis as compared to the Ni/S ratio of 30 during catalyst preparation. This occurrence can be reasonably explained by the enrichment of S on the catalyst surface. Conversely, the Ni/S ratio of 28.4 from elemental analysis and ICP-OES results, which is slighter lower than 30, may be attributed to the difference of experimental measurement methods. Hereinto, the Ni contents in the samples were determined by the ICP-OES measurement using aqua regia digestion of catalyst samples. The S contents were determined by the elemental analysis using quantification of SO₂ emission which were generated from burning the catalysts in oxygen-rich environment.

Furthermore, the measurement process of the Ni@C-S_{1/30} catalyst was repeated three times to identify the primary source of deviation. As shown in Table R2, the deviation mainly exists in the S elemental analysis (mean absolute deviation: 0.24-5.59%). Therefore, we believe that the deviation of the Ni/S ratio is caused by the low S content in the Ni@C-S_{1/30} catalyst and the unavoidable measurement error of the elemental analysis instrument.

Table R2 |The elemental analysis and ICP-OES results of the Ni@C-S_{1/30} catalyst obtained from three times re-measured.

Entry	S ^a (wt%)	Mean absolute deviation of S content ^b (%)	Ni ^c (wt%)	Mean absolute deviation of Ni content ^d (%)	Ni/S ^e
1st	1.18	5.59	68.62	3.46	32.0
2nd	1.32	5.35	65.25	1.63	27.3
3th	1.26	0.24	65.11	1.83	28.6

^aS contents in the catalysts were determined by elemental analysis.

^bMean absolute deviations of S content were calculated based on the average of elemental analysis results.

^cNi contents in the catalysts were determined by ICP-OES analysis.

^dMean absolute deviations of Ni content were calculated based on the average of ICP-OES results.

^eThe Ni/S atomic ratios were calculated from elemental analysis and ICP-OES results.

7. In this manuscript, the S content affects the catalytic performance, so it is very important to determine this Ni/S ratios, especially the Ni/S ratio of the surface (5.7 at.% by XPS analysis), and they may affect the validity of DFT computational model. Please check.

We thank the referee for his/her insightful suggestion. As the referee pointed out, the Ni/S ratios detected by XPS analysis should be taken into account when building the DFT calculation models. It is well known that XPS is a surface sensitive quantitative analysis technique, which identifies chemical species and quantifies their elemental compositions and chemical states, with an average analysis depth of 1–10 nm (*Prog. Mater. Sci.* **107**, 100591 (2020)). Generally, the practically attainable detection limits are in the range of 0.1 to 1 at.%. An empirical approach is applied to calculate the atomic percentage on the surface layer using XPS peak areas under the main core-level lines of all elements present in the sample, based on the normalized corrected signals. For a homogenous sample containing n elements, the molar concentration x_i of element i is formularized as follows:

$$x_i = \frac{A_i/s_i}{\sum_{j=1}^n (A_j/s_j)}$$

in which A_i is the area under the corresponding core-level peak, and s_i is the relative sensitivity factor (RSF).

Notwithstanding, the measurements by XPS are generally a precise process yet may be inaccurate for elemental quantification in certain cases. Statistical uncertainties in elemental quantification are ± 1 % or lower for large spectral peaks and greater for small peaks or noisy spectra (*Can. J. Chem. Eng.* **97**, 2588-2593 (2019)). Systematic uncertainties reach 50 % in the worst case and contribute the most to inaccuracy, particularly due to applying an incorrect model. The peak area ratio (normalized by sensitivity factors) is the most common quantification method that the sample is assumed homogeneous, which is often erroneous for samples exposed to air because adventitious carbon contaminates the surface and/or a surface oxide layer forms. The sample's surface, in fact, seems significantly rough, with an extremely uneven distribution of surface elements. Ultimately, statistical uncertainties are related not only to the concentration of atoms, but also to the mean free path of the photoelectrons, the surface finish of the sample, the chemical state of the element, the intensity of the X-ray source and the state of the instrument (*Mater. Chem. Front.* **5**, 7931 (2021)). Sometimes, XPS analysis cannot be treated as a semi-quantitative method used for determine the elemental composition in the

target materials. Based on our special sample morphology, high-resolution TEM (HRTEM) images demonstrate that the Ni NPs are well encapsulated, but not tightly, by the graphitized carbon layers with ~3 nm thickness (see Supplementary Fig. 4-6). In particular, the low-content S element is predominantly distributed around the Ni NPs, and the proportion of nickel content and sulfur content varies widely. These realistic factors result in the situation that XPS spectra are inaccurate for reflecting the true Ni/S ratios on the surface of S-doped Ni@C-S_x catalysts.

For this reason, the XPS analysis results cannot be the only reference. The establishment of DFT computational model needs to refer to the experimental results of aqueous EtOH upgrading (Fig. 2a) and *n*-butanal adsorption (Fig. 3b-f).

Since the sulfur species on the Ni surface can affect the adsorption of *n*-butanal, we calculated the adsorption energy of *n*-butanal on S-free and S-doped Ni surfaces to explore the structure–activity relationship. The Ni (111) slab model consisting of three layers (3×3 supercell and 108 atoms in total) was firstly constructed based on the optimized bulk configuration to simulate the S-free Ni surface. Then, different numbers of S atoms were added on the top of Ni (111) slab model (consisting of 36 atoms) to simulate the S-doped Ni surface with different sulfur coverage. Here, the sulfur coverage was the atomic percentage of S/Ni_{top layer}.

The Ni/S ratio of 5.7/1 corresponds to the sulfur coverage of $1/(5.7/1)*100\% = 17.5\%$. The models in Fig. 4a are in agreement with the experimental results (Fig. 2a and Fig. 3b-f) and XPS analysis (Supplementary Table 1 and Table R3). When the sulfur coverage is lower than 17.5%, the high adsorption energy of *n*-butanal is accompanied by the increase of LAS selectivity; when the sulfur coverage is higher than 17.5%, the adsorption energy of *n*-butanal is gradually reduced, accompanied by the decrease of LAS selectivity. Therefore, we believe that these models can effectively reflect the surface structure of Ni@C-S_x catalysts.

On this basis, the models capable of representing the typical catalysts were selected from a series of models for the calculation of Gibbs free energies. The Ni (111) surface covered with 0% sulfur was chosen as the model of S-free Ni@C-S₀ catalyst. The Ni (111) surface covered with 14% sulfur, which is the closest to 17.5%, was chosen as the model of optimized Ni@C-S_{1/30} catalyst. In addition, the Ni (111) surface covered with 25% sulfur, which corresponds to the adsorption energy of *n*-butanal with sharp decrease, was chosen as the model of Ni@C-S_{1/25} catalyst. These models were used to calculate the Gibbs free energies for the RLS of ethanol dehydrogenation and C–C bond cleavage. As shown in Fig. 4b-c, the results of the activation energies are also consistent with our experimental results

(Fig. 2a), further confirming that the established DFT computational model can availablely reflect the structure–activity relationship in this work.

Table R3 |The XPS elemental compositions of Ni@C-S_x catalysts.

Sample	Elemental composition (at%)			
	Ni	S	C	O
Ni@C-S _x	9.11	-	84.55	6.34
Ni@C-S _{1/30}	8.15	1.43	82.81	7.61
Ni@C-S _{1/25}	9.12	1.70	78.73	10.45

8. Lines 132-133, the authors mentioned that “The deconvoluted peaks of S 2p spectra can be assigned to Ni–S (162.3 and 163.7 eV)”. Is the Ni–S Sulfide or Sulfate?

We thank the referee for his/her valuable question. According to reference 29 (*Nat. Commun.* **8**, 14580 (2017)) in our manuscript, the S 2p XPS peaks located at 162.3 and 163.7 eV are attributed to sulfide.

9. Lines 180-183, the authors mentioned that “Among the two phases, the upper oil phase consists mainly of 1-butanol and LAS (e.g., 2-ethyl-1-butanol, 1-hexanol and 2-ethyl-1-hexanol), together with very small quantities of alkanes and aldehydes, whereas the unreacted EtOH and water are primarily detected in the aqueous phase”. The catalysts stay in which phase in the reaction?

We thank the referee for his/her thoughtful question. As shown in Fig. R4, the majority of catalysts are distributed in the organic phase, with only small amounts being adsorbed by the magnetic stirring bar when the reaction system remains stagnant in the oil–water layering. Under vigorous stirring (1,000 rpm), the organic phase and aqueous phases would form a temporarily stable mixture (i.e., an emulsion). Hence, we rationally hypothesize that the catalysts in the reaction could come into full contact with both the organic phase and aqueous phases during vigorous stirring.

Fig. R4 | The phase behavior of the reaction product under (a) stirring condition and (b) static condition.

10. Line 229, please explain the rapid-swift rate in the sentence of “...Taken together with the rapid-swift rate in C–C bond formation...”.

We thank the referee for bringing this to our attention. To underscore the faster rate of C–C bond formation (Aldol condensation) (see Supplementary Table 11) compared to that of dehydrogenation and hydrogenation in Guerbet condensation (see Supplementary Table 10), we used the term “rapid-swift”. To enhance the readability, we have revised the sentence to “...Taken together with the rapid-swift catalysis process of forming C–C bond...”.

11. Line 308, why the Ni (111) surfaces covered with 0%, 14% and 25% sulfur was selected for the DFT computational model?

We thank the referee for his/her insightful question. Given that the active phase of Ni in the catalyst is predominantly face-centered cubic (fcc) Ni, with the (111) plane being the main crystal face for fcc Ni, we have consulted previous studies (*ACS Catal.* **12**, 11573-11585 (2020); *Appl. Catal. B.* **321**, 122048 (2023)) to build a Ni (111) slab model, and then added various numbers of S atoms onto the top of the Ni (111) slab model to simulate the Ni surface with different sulfur coverage. As discussed in the response to Comment 7, we comprehensively analyzed the experimental results (refer to Fig. 2a and Fig. 3b-f) as well as the Ni/S ratio obtained by XPS analysis (see Supplementary Table 1). Ultimately, we selected Ni (111) surfaces covered with 0%, 14% and 25% sulfur as the theoretical models of the Ni@C-S₀, Ni@C-S_{1/30} and Ni@C-S_{1/25} catalysts, correspondingly.

12. Lines 358-360 and Supplementary Fig. 24, the authors mentioned that “...the deconvoluted XPS peaks assigned to S–Ni and S element distributed around the Ni NPs can also be observed over the Ni@C-DMSO_{0.006}...”. Are the S species in this case Sulfides or Sulfates? Their status should be clarified to justify the explanations in the discussion.

We thank the referee for his/her helpful suggestion. The reference 50 (*ACS Catal.* **11**, 9204-9209 (2021)) in the manuscript has identified that the S 2p peak with a binding energy of 162.6 eV can be assigned to disulfide on the recovered Ru/C following co-reaction with DMSO. The authors have compared the S 2p XPS spectra with those of bulk RuS₂ in the literature, confirming that the DMSO interacts with Ru in the form of sulfides. Similarly, we believe that the broad peak at ~162.6 eV should be assigned to the Ni-S sulfide. Additionally, it is recommended to assign the S 2p peak at the binding energy of ~169.0 eV to the sulfur oxide (*Nat. Commun.* **12**, 3881 (2021)), potentially due to the residual DMSO. To prove this assignment, we prepared the DMSO/Ni@C catalyst by immersing the Ni@C catalyst in 10% DMSO and subsequently drying it under Ar atmosphere at room temperature. Further XPS analysis was performed to demonstrate that the sulfur oxide originated from the remaining DMSO on the surface of the Ni@C-DMSO_{0.006} catalyst (Fig. R5).

To sum up, we believe that the Ni-S species in catalysts belong to Ni-S sulfide, which originate from the interaction between Ni surface and DMSO.

Fig. R5 | High-resolution S 2p XPS spectra for Ni@C-DMSO_{0.006} catalyst and DMSO/Ni@C.

13. Lines 405-406, the authors mentioned that “...the dried gels were carbonized at 550 °C for 2 h with a 5 °C min⁻¹ heating rate under N₂ atmosphere to obtain the catalysts.” in the synthesis of Ni@C-S_x catalysts. But “... the fully dried solid gels were annealed at 600 °C for 4 h with a heating rate of 5 °C min⁻¹ under N₂ atmosphere” in the synthesis of NiSn@C

catalyst. And “... the dried gel was carbonized at 500 °C for 2 h with a heating rate of 10 °C min⁻¹ under N₂ atmosphere.” in the synthesis of Ni@NC catalyst. Is there anything special about the choices of heat treatment parameters here to achieve a certain goal?

We thank the referee for his/her insightful comment. The heat treatment parameters for preparing NiSn@C and Ni@NC catalysts are referenced directly from the optimized carbonization parameters in our previous works (*Energy Convers. Manag.* **249**, 114822 (2021); *Chem. Eng. J.* **453**, 139583 (2023)). For NiSn@C catalysts, by changing the carbonization temperature, the Sn⁰/Ni⁰ ratio could be controlled effectively, thereby adjusting the carbon chain step-growth ability. For Ni@NC catalysts, changing the carbonization temperature could regulate the N species and N contents in the N-doped carbon layers surrounding Ni nanoparticles, which play a key role in inhibiting the formation of C₁ byproducts and facilitating the coupling reaction to produce more alcohols.

We previously invested a significant amount of effort in optimizing the heat treatment parameters to enhance the catalytic performance. Moreover, the latest published research (*Appl. Catal. B.* **321**, 122048 (2023)) has highlighted the significance of heat treatment parameters as primary process parameters in catalyst preparation that affect catalytic properties. Clearly, the carbonization temperature exerts significant influences over the particle size of Ni nanoparticles, the graphitization extent of carbon layers coating on the Ni nanoparticles and the reduction degree of Ni species. Given this complexity of the relationship, the influence of carbonization temperature was also inspected for determining the appropriate carbonization temperature of Ni@C-S_{1/30} catalysts in Fig. R6-7 and Table R4. In our investigation, we found that Ni species can usually be fully reduced to metallic Ni at 500 °C, but the catalyst surface usually harbors many Ni²⁺ species (XPS analysis results of reference 13, 25 and 27 in the manuscript). Increasing the carbonization temperature to 600 °C can effectively reduce the content of Ni²⁺ species on the catalyst surface. Nevertheless, too high carbonization temperature usually leads to the increase of Ni nanoparticle sizes and the tight wrapping of Ni nanoparticles by highly graphitized carbon layers, thus leading to a decrease of catalytic activity. Taken together, we have selected 550 °C as the appropriate carbonization temperature in this work, in order to ensure the exposure of more active Ni sites on the catalyst surface for achieving superior catalytic activity in upgrading aqueous EtOH to LAS.

Fig. R6 | XRD patterns of the Ni@C-S_{1/30} catalysts with different carbonization temperatures.

Fig. R7 | Catalytic performance of aqueous EtOH coupling over Ni@C-S_{1/30} catalysts with different carbonization temperatures.

Table R4 | Catalytic performance of aqueous EtOH coupling over Ni@C-S_{1/30} catalysts with different carbonization temperatures.^a

Carbonization temperature (°C)	Conversion (%)	Carbon balance (%)	Selectivity (C%)						
			C ₄ OH	C ₆ OH	C ₈₊ OH	Ald	L-CH	Gas	Other
500	64.1	87.5	22.4	28.0	38.5	1.1	1.0	7.0	2.0
550	76.5	84.8	21.3	27.5	43.0	1.4	0.9	3.5	2.4
600	56.3	85.5	37.2	27.7	27.3	3.1	1.3	2.1	1.3

^aReaction conditions: 180 °C, 12 h, catalyst (0.3 g), NaOH (21.6 mmol), 50.0 wt% aqueous EtOH (108.0 mmol).

14. Line 521, the citation format needs to be improved.

We thank the referee for bringing this to our attention. We have now modified the citation format in the revised manuscript.

15. *Is it Ni@C-S_{1/30} in the XPS spectra for Ni@C-S_x catalysts Supplementary Fig. 15?*

We thank the referee for pointing out this issue. We have corrected the Ni@C-S_x to Ni@C-S_{1/30} in the revised Supplementary Fig. 16.

16. *Please clarify the reaction conditions of “under optimized conditions” for Supplementary Fig. 27 in SI.*

We thank the referee for his/her thoughtful suggestion. According to the suggestion, we have now added the detailed reaction conditions for Supplementary Fig. 27.

17. *Page 41, under the Supplementary Table 6, it may be “Conversion × Selectivity”.*

We thank the referee for bringing this to our attention. According to the suggestion, we have made the corresponding modification in Supplementary Table 6.

18. *Page 45, under the Supplementary Table 10, it may be “Calculated from the results in Supplement Table S 8 and Table S9”.*

We thank the referee for bringing this to our attention. According to the suggestion, we have made the corresponding modification in Supplementary Table 10.

19. *Page 45, under the Supplementary Table 10, why the hydrogenation rate was much higher than the dehydrogenation rate?*

We thank the referee for his/her thoughtful question. According to previous research studies (*Nat. Rev. Chem.* **3**, 223-249 (2019); *Catal. Sci. Technol.* **5**, 3876 (2015)), dehydrogenation of a primary alcohol is highly endothermic ($\Delta H^\circ = 68.5 \text{ kJ mol}_{\text{EtOH}}^{-1}$ and $69.4 \text{ kJ mol}_{n\text{-butanol}}^{-1}$, respectively). Hence, it becomes thermodynamically favorable ($\Delta G < 0$) only above 320 °C. On the other hand, complete hydrogenation of 2-butenal to 1-butanol is highly exothermic ($\Delta H^\circ = -171 \text{ kJ mol}^{-1}$). This difference in enthalpy specifies

that the 2-butenal hydrogenation is more thermodynamically favorable. Therefore, it is reasonable to attribute the higher hydrogenation rate of 2-butenal, compared to the dehydrogenation rate of EtOH, to the thermodynamic advantage.

Reviewer #2:

In this manuscript, the authors developed a catalyst with controllably exposed Ni catalytic sites on the surface through sulfur modification. This modification effectively stabilizes and enriches aldehyde intermediates, leading to a significant improvement in the direct-growth probability of LAS production. While the study includes routine work and theoretical insights from DFT calculations about the varying sulfur coverages' role in 1-butanol adsorption, the experimental demonstration of the structure-activity relationship remains unclear.

We really appreciate the referee's positive evaluation for the improvement in the direct-growth probability of LAS production by our work, and are very grateful to the referee for his/her insightful suggestions to help us significantly improve the quality of our manuscript. Specifically, we thank the referee for raising the concerns regarding the structure-activity relationship in this work. We have made point-by-point responses to the referee's comments as listed below. We sincerely hope that our careful revisions have satisfactorily addressed the referee's concerns (see below).

Furthermore, the authors' previous work (Applied Catalysis B: Environmental, 2023, 321, 122048) has already explored the modification of aldehyde intermediate adsorption for C-C bond cleavage over Ni catalysts, which may reduce the novelty of the current study.

We really appreciate the referee's attention to our previous work. In our previous work (*Appl. Catal. B* **321**, 122048 (2023)), our objective was to inhibit C-C bond cleavage on Ni catalysts, as a prerequisite for enhancing ethanol conversion and improving the alcohol product selectivity. Specifically, we modified the electron density surrounding the Ni active sites through Sn doping, which resulted in the weakened adsorption of intermediate aldehydes, and consequently, inhibiting C-C bond cleavage on the Ni sites. The optimized Sn-Ni/CS catalyst can achieve a high alcohol selectivity (86.4%) with 60% ethanol conversion. Nonetheless, the main alcohol product is *n*-butanol (~50%) and the corresponding alcohol distribution still succumbs to the step-growth model.

The previous literatures (*ACS Sustain. Chem. Eng.* **10**, 3466-3476 (2022); *Chem. Commun.* **55**, 10420-10423 (2019)) have indicated that aldehyde intermediates should be enriched on the catalyst surface for enhancing the LAS (C₆₊ alcohol) selectivity. As such, the key to achieving a high LAS yield relies on preserving the strong adsorption sites while inhibiting the C-C bond cleavage on the catalysts. However, the Sn doping strategy cannot fulfil this criterion because of its insufficient strong adsorption sites.

Our key innovation in this work is the introduction of a small amount of sulfur to occupy the partial threefold hollow sites on Ni. This strategy aims to hinder the C–C bond cleavage while minimally weakening the strong adsorption of aldehyde intermediates. As a result, the direct-growth probability toward LAS production is dramatically improved.

Additionally, prior research (*ACS Sustain. Chem. Eng.* **10**, 3466–3476 (2022)) and our own experimentation (Fig. 2a) have indicated that the C–C bond cleavage on Ni sites can reduce the LAS selectivity. For this reason, we aim to systematically examine the modification of aldehyde intermediate adsorption on the Ni@C-S_x catalysts for C–C bond cleavage in this work.

The detailed comments are shown below:

1. The authors have proposed two routes for ethanol coupling, namely a step-growth route and a direct-growth route. In these pathways, 1-butyraldehyde serves as the key intermediate and undergoes either hydrogenation or condensation reactions depending on the concentration of the intermediate aldehyde. Therefore, I believe it is not appropriate to attribute the high yield of long-chain alcohols to differences in routes. Additionally, butanol is typically regarded as a fully hydrogenated molecule that presents greater difficulty in dehydrogenation, so the formation of long-chain alcohols is still mainly through the condensation of intermediate aldehydes. The authors should consider this point carefully.

We thank the referee for his/her insightful suggestion. We agree that the high yield of long-chain alcohols is mainly due to the condensation of intermediate aldehydes on exposed Ni sites. However, it has been demonstrated in previous work (*ACS Catal.* **11**, 14, 8521–8526 (2021)) that although *n*-butanol is more challenging to dehydrogenate than ethanol, it can still be successfully dehydrogenated by the metal active sites at 170 °C and subsequently upgraded to 2-ethylhexanol through the Guerbet reaction. Furthermore, we have experimentally verified that *n*-butanol can be dehydrogenated to *n*-butanal and upgraded to long-chain alcohols over Ni@C-S_{1/30} (Table R5), albeit with a lower conversion than ethanol. Based on this evidence, it is not possible to fully attribute the formation of long-chain alcohols to the condensation of intermediate aldehydes. Therefore, we suggest that both the direct-growth route and the step-growth route should occur simultaneously in the reaction process, except that the reaction process is more inclined toward the direct-growth route to the production of long-chain alcohols.

Table R5 | Catalytic performance of Ni@C-S_{1/30} catalyst in the coupling of EtOH and *n*-butanol.

Reactant	Conversion (%)	Carbon balance (%)	Selectivity (C%)						
			C ₄ OH	C ₆ OH	C ₈₊ OH	Ald	L-CH	Gas	Other
EtOH	76.1	85.9	21.6	26.6	46.2	0.7	0.9	3.2	0.8
n -butanol	51.6	87.8	-	-	94.2	-	0.7	2.6	2.5

^aReaction conditions: 180 °C, 12 h, Ni@C-S_{1/30} catalyst (0.3 g), NaOH (21.6 mmol), 50.0 wt% aqueous alcohol (108.0 mmol).

2. How does Ni maintain a high dispersion at such a high loading (~65%) in Ni@C-S_x catalysts. And is this affected by the support or sulfur modification?

We thank the referee for his/her thoughtful question. Based on the XRD patterns (Supplementary Fig. 2) and TEM images (Supplementary Fig. 4-5) of typical Ni@C-S_x catalysts, the S-free Ni@C-S₀ catalyst displays comparable Ni dispersion to the S-modified Ni@C-S_x catalysts. Therefore, the high dispersion at such a high loading (~65%) in Ni@C-S_x catalysts can be attributed to the special carbon support (carbon layers formed in-situ during carbonization). According to the literature, the Ni@C-S_x catalysts could be categorized as “chainmail catalyst” (*Adv. Mater.* **29**, 1606967 (2017); *Angew. Chem.* **127**, 2128–2132 (2015)). After carbonization in inert atmosphere, the organic ligands complexed with metals ion could be transformed into graphitized carbon shells that surround metal nanoparticles through *in situ* reduction and growth, eventually forming unique chainmail catalysts. The stable chainmail carbon layers could encapsulate and protect the inner small-sized metal nanoparticles from aggregation, thereby achieving high dispersion and high Ni loading.

3. In Fig. 2a, the catalysts with different S/Ni molar ratios exhibited diverse reaction activities, with the Ni@C-S_{1/30} catalyst having the best ethanol conversion and long chain alcohol selectivity. It can be noted that the Ni@C-S_{1/25} catalyst has higher conversion and higher butanol selectivity than the sulfur-free catalyst. However, in the *in situ* DRIFTS spectra of Figure 3d-f, the Ni@C-S_{1/25} catalyst exhibited a completely different spectral signals. In Figure 3f, we even do not observe the obvious signal of butanal over 50 °C. More explanations are required.

We thank the referee for his/her insightful suggestion. In this work, we carried out *in situ* DRIFTS to prove that the Ni sites, strongly adsorbing aldehydes but inert for side reactions,

are critical to LAS production. In Figure 3d-f, the adsorption strengths of the catalysts for *n*-butanal are $\text{Ni@C-S}_0 \cong \text{Ni@C-S}_{1/30} > \text{Ni@C-S}_{1/25}$. In addition, as some *n*-butanal would dissociate over Ni@C-S_0 , the abilities of the catalysts for enriching *n*-butanal are $\text{Ni@C-S}_{1/30} > \text{Ni@C-S}_0 > \text{Ni@C-S}_{1/25}$. The *in situ* DRIFTS results are consistent with the LAS selectivity in the alcohol products (Fig. 2 and Supplementary Fig. 9). The chain growth probabilities (α) of the catalysts are $\text{Ni@C-S}_{1/30}$ (0.46) $>$ Ni@C-S_0 (0.42) $>$ $\text{Ni@C-S}_{1/25}$ (0.31), indicating that the chain growth capacity of the $\text{Ni@C-S}_{1/25}$ catalyst is the lowest among the catalysts. For this reason, the main product for the $\text{Ni@C-S}_{1/25}$ catalyst is butanol rather than LAS.

As for the catalysis results, the $\text{Ni@C-S}_{1/25}$ catalyst displayed a higher conversion than the S-free catalyst. This is in line with our previous study (*Chem. Eng. J.* **453**, 139583 (2023)), which highlighted that suppressing of the C-C bond cleavage in aldehydes can enhance the dehydrogenation equilibrium of ethanol. The dissociation of aldehydes results in the production of H_2 , CO and CO_2 , which would occupy the Ni sites and alter the dehydrogenation equilibrium. In particular, CO undergoes a water-gas shift reaction over the Ni sites (*ACS Catal.* **7**, 11, 7600–7609 (2017)), leading to production of significant amounts of H_2 and CO_2 . Both the presence of high-pressure H_2 (*Catal. Sci. Technol.* **5**, 3876 (2015)) and the consumption of NaOH (*J. Mol. Catal. A: Chem.* **212**, 65 (2004)) neutralizing with CO_2 would suppress the dehydrogenation of ethanol.

Furthermore, we have conducted the verification experiments to substantiate the above judgments. As shown in Fig. R8 and Table. R6, the introduction of CO and H_2 into the reaction system led to the decrease of ethanol conversion and LAS selectivity. Especially, CO had the more pronounced inhibition effect on ethanol dehydrogenation. For this reason, the S-free Ni@C-S_0 catalyst, which inclined to C–C bond cleavage of aldehydes, exhibits a lower ethanol conversion than that of the $\text{Ni@C-S}_{1/25}$ catalyst.

Fig R8 | Comparison of ethanol coupling over the Ni@C-S_{1/30} catalyst under different reaction atmospheres.

Table R6 | Comparison of ethanol coupling over the Ni@C-S_{1/30} catalyst under different reaction atmospheres.

Reaction atmosphere	Conversion (%)	Carbon balance (%)	Selectivity (C%)						
			C ₄ OH	C ₆ OH	C ₈₊ OH	Ald	L-CH	Gas	Other
0.1 MPa H ₂	76.1	85.9	21.6	26.6	46.2	0.7	0.9	3.2	0.8
0.5 MPa 47.5% H ₂ /47.5% CO/5% N ₂	44.5	84.6	37.1	29.8	29.4	-	-	3.1	0.5
0.5 MPa 100% CO	26.9	87.6	38.3	29.3	26.5	-	-	4.4	1.5

^aReaction conditions: 180 °C, 12 h, Ni@C-S_{1/30} catalyst (0.3 g), NaOH (21.6 mmol), 50.0 wt% aqueous alcohol (108.0 mmol).

4. *The author's algorithm for C-C bond formation rate may present a potential issue. In my opinion, it would be counted in terms of the product, not the ethanol of the reaction, as not all products are exclusively coupling products.*

We thank the referee for his/her thoughtful suggestion. To enhance the comprehension of the impact of sulfur on tuning catalytic performance, we have performed the simplified model reactions, such as EtOH dehydrogenation, 2-butanal hydrogenation and *n*-butanal condensation, allowing us to uncover fundamental factors that contribute to the high-yield production of LAS in aqueous EtOH coupling. It must be emphasized here that the C–C bond formation rate pertains to the condensation rate of *n*-butanal in the aldol condensation reaction. Since *n*-butanal can be completely condensed to 2-ethyl-2-hexenal in the presence of NaOH, we have used the mole number of reacted *n*-butanal to calculate the condensation rate, instead of the complicated product distribution. We have also made appropriate modification on the C–C bond formation rate equation in the Supplementary Information.

5. *The authors use 50 wt% aqueous ethanol solution as a feedstock, why not feed higher purity ethanol in this case? Additionally, would water potentially have any role in the catalytic reaction or catalyst?*

We thank the referee for his/her valuable comment. In this work, a 50.0 wt% ethanol aqueous solution was adopted to mimic bioethanol, which can be widely obtained from biomass fermentation followed by distillation. The coupling reaction of anhydrous ethanol

was also conducted for assessing the catalytic activity, and the corresponding results are presented in Fig. R9 and Table R7. As suggested, the experimental results indicate that the existence of water could enhance the aqueous ethanol coupling, resulting in improved LAS selectivity and ethanol conversion. Unexpectedly, conducting the reaction in anhydrous ethanol leads to the formation of viscous solids in the solution, causing the reaction mixture to thicken. This undesirable interaction between the catalyst and reactants causes further decrease in the carbon balance from 85.9% to 78.1%. After the reaction, we collected the viscous solids and analyzed them through XRD. The XRD analysis reveals that viscous solids mainly consist of $\text{Na}_2\text{CO}_3 \cdot \text{H}_2\text{O}$ (Fig. R10), which was derived from the consumption of NaOH by small amounts of CO_2 byproducts. These results indicate that the presence of water can promote the solubility of $\text{Na}_2\text{CO}_3 \cdot \text{H}_2\text{O}$ in the reaction medium, thereby guaranteeing the accurate evaluation of catalyst performance.

Moreover, the role of water in promoting the conversion of ethanol and the production of LAS may also originate from other mechanisms. Combined with the literature (*Green Chem.* **23**, 430-439, 11 (2021); *J. Phys. Chem. C.* **124**, 9385-9393 (2020)), we know that the double H-atom exchange reactions could lower the activation energy of alcohol dehydrogenation. Additionally, the adsorption of water on catalyst may facilitate a pathway shifting effect that diverts the reactive flux toward the production of aldehydes.

However, the influence of water on the ethanol coupling has yet to be demonstrated and remains unclear. Conducting a comprehensive investigation is highly recommended. In our forthcoming work, we will endeavor to provide clarification on the catalytic pathways and document them systematically.

Fig. R9 | Comparison of ethanol coupling over the Ni@C-S_{1/30} catalyst with 50.0% aqueous ethanol and 99.7% anhydrous ethanol.

Table R7 | Comparison of ethanol coupling over the Ni@C-S_{1/30} catalyst with 50.0% aqueous ethanol and 99.7% anhydrous ethanol.^a

Ethanol concentration (%)	Conversion (%)	Carbon balance (%)	Selectivity (C%)						
			C ₄ OH	C ₆ OH	C ₈₊ OH	Ald	L-CH	Gas	Other
50.0	76.1	85.9	21.6	26.6	46.2	0.7	0.9	3.2	0.8
99.7 ^b	72.0	78.1	44.4	27.8	20.5	1.0	0.7	5.3	0.4

^aReaction conditions: 180 °C, 12 h, Ni@C-S_{1/30} catalyst (0.3 g), NaOH (21.6 mmol), EtOH (108.0 mmol).

^bAnhydrous ethanol (99.7%) was purchased from Macklin Biochemical Co., Ltd., China and used without further purification.

Fig. R10 | XRD patterns of the viscous solids generated from the anhydrous ethanol reaction system.

6. From the DFT calculation, the adsorption energy of butanal on sulfur-free Ni@C and sulfur-rich Ni@C catalysts are pretty close. How to explain the large ethanol conversion difference between Ni@C-S_{1/30} catalyst (76.5 %) and sulfur-free Ni@C (only 51.8 %) at 180 °C.

We thank the referee for his/her helpful comment. As mentioned in our response to Comment 3, the severe dissociation of aldehydes on the S-free Ni@C catalyst impedes the ethanol dehydrogenation. Consequently, the Ni@C-S_{1/30} catalyst, which is inert to the C–C bond cleavage of aldehydes, displays a higher ethanol conversion compared to the S-free Ni@C catalyst.

7. It is noted that the carbon balance of the different catalysts is below 90%. How about carbon deposition of the catalyst, please give the TG profiles after the reaction.

We thank the referee for his/her valuable suggestion. The carbon balance data were calculated based on the collected gas and liquid products, using the following equations in Supplementary Information:

$$\text{Carbon balance (C-mol\%)} = \frac{\sum \text{Moles of carbon in all detected products}}{\text{Moles of carbon in feedstock}} \times 100\%$$

Owing to the existence of the high proportion of organic compounds in products, specifically volatile ethanol and oleophilic long-chain alcohols, it is very difficult to collect all the organic compounds from the reaction system after reaction. We infer that the insufficient carbon balance is attributable to small amounts of liquid products adsorbed on the carbon-wrapped Ni catalyst and the unavoidable loss of ethanol volatilized in the depressurization and product collection processes.

Furthermore, we have conducted a blank experiment without a catalyst. The blank experiment yielded no gas products, and only ethanol was detected in the collected liquid product. The carbon balance showed a 7.7% reduction, which can be attributed to the carbon loss of ethanol volatilization (Table R8). Additionally, we washed the used Ni@C-S_{1/30} catalyst with methanol under ultrasonic conditions, and quantitatively analyzed the resulting products in methanol. It was found that the organic products and residual ethanol on the used catalyst were responsible for a carbon loss of 4.2% (Table R9).

In contrast to the residual liquid products trapped on the catalyst surface, carbon deposition usually denotes to the carbon or coke generated via the decomposition and polymerization of carbon-containing species attached to the catalyst surface (*Chem. Eng. J.* **322**, 339-345, 11 (2017); *Appl. Catal. B.* **239**, 502-512 (2018)). Consequently, the used Ni@C-S_{1/30} catalyst was cleaned of residues, centrifuged, dried and then subjected to TG analysis to evaluate the degree of carbon deposition. As shown in Fig. R11, the used Ni@C-S_{1/30} catalyst that had been used for a typical test run displays a minimal weight loss, comparable to that of the fresh catalyst. These findings indicate that there is almost no carbon or coke on the used Ni@C-S_{1/30} catalyst. With further analysis of its contribution to the carbon loss, we think that the carbon deposition on the spend catalyst has a negligible effect on the carbon balance.

Taken together, the low carbon balance should be mainly ascribed to the residual liquid products adsorbed on the catalyst and the unavoidable carbon losses from ethanol volatilized during the depressurization and product collection processes.

Table R8 | Typical aqueous EtOH coupling and corresponding blank experiments over Ni@C-S_x catalysts.

	Conversion (%)	Carbon balance (%)	Selectivity (C%)						
			C ₄ OH	C ₆ OH	C ₈₊ OH	Ald	L-CH	Gas	Other
Typical experiment ^a	76.1	85.9	21.6	26.6	46.2	0.7	0.9	3.2	0.8
blank ^b	7.7	92.3	-	-	-	-	-	-	-

^aReaction conditions: 180 °C, 12 h, Ni@C-S_{1/30} catalyst (0.3 g), NaOH (21.6 mmol), 50.0 wt% aqueous EtOH (108.0 mmol).

^bReaction conditions: 180 °C, 12 h, NaOH (21.6 mmol), 50.0 wt% aqueous EtOH (108.0 mmol).

Table R9 | Residual liquid product composition and its contribution to carbon loss.

	Contribution to carbon loss (%) ^b	Composition (C%)							
		EtOH	C ₄ OH	C ₆ OH	C ₈₊ OH	Ald	L-CH	Gas	Other
Residual liquid products ^a	4.2	23.7	20.5	23.0	32.8	-	-	-	-

^aThe used Ni@C-S_{1/30} catalyst was separated by centrifugation and then put into 5 g methanol ultrasonic washing for 30 min. Finally, after centrifugation, the used methanol solution was analyzed by gas chromatography.

^bThe contribution of liquid products to carbon loss is evaluated using the equation of carbon balance.

Fig. R11 | TG profiles of fresh/used Ni@C-S_{1/30} catalyst.

TG analysis: the thermogravimetric data were recorded using a synchronous TG thermal analyzer (Mettler Toledo, Switzerland) with a scan range from room temperature to 900 °C in airflow and a heating rate of 10 °C/min.

8. *The authors mentioned, "the introduction of low-content sulfur can not only effectively reduce the number of active sites for C–C bond cleavage and thus increase the temperature required for aldehyde dissociation, but also retain abundant adsorption sites for aldehydes simultaneously". If possible, it is recommended that the number of active sites is quantified to facilitate a better structure-activity relationship.*

We thank the referee for his/her insightful suggestion. In our manuscript, we refer the active Ni sites to as those surface Ni atoms which are exposed and capable of adsorbing aldehydes. According to relevant literatures (*ACS Appl. Mater. Interfaces* **13**, 28334–28347 (2021); *Appl. Catal. B.* **322**, 122138 (2023); *Angew. Chem. Int. Ed.* **55**, 1080–1084 (2016)), the surface Ni atoms in the Ni-based catalysts can usually be quantified by pulsed N₂O titration, pulsed CO chemisorption and pulsed H₂ chemisorption. As illustrated in Supplementary Fig. 8, the *in situ* DRIFTS spectra clearly confirm the presence of sulfur covering the Ni surface with a lower affinity for CO in the S-doped Ni@C-S_x catalysts, leading to the formation Ni(CO)₄ on the catalyst surface. Since the adsorption mode of CO on the Ni@C-S_x catalysts differs from that of the S-free Ni@C-S₀ catalyst, we excluded the use of pulsed CO chemisorption method, and initially aimed to titrate the surface Ni atoms using pulsed H₂ chemisorption and pulsed N₂O titration.

As shown in Table R10, the surface atomic numbers measured by pulsed H₂ chemisorption and pulsed N₂O titration are quite different (26.9 vs. 3.4 μmol·g⁻¹). We suggest that the interaction between Ni and graphitized carbon layers makes the H₂ adsorption electronically less favorable (*Appl. Surf. Sci.* **447**, 254–260 (2018)), resulting in a low H₂ dissociation adsorption capacity on the Ni@C-S₀ catalyst. In contrast, N₂O titration is a method based on surface oxidation and is commonly used for the determination of metals with weak H₂ dissociation adsorption capacity (*Appl. Catal. B.* **297**, 120398 (2021)), and has been shown to be an effective method for measuring the number of Ni atoms on the surface (*Environ. Sci. Technol.* **53**, 10379–10386 (2019); *Appl. Catal. A-GEN.* **503**, 34–42 (2015)). Therefore, we finally chose the pulsed N₂O titration method to further quantify the surface Ni atoms of Ni@C-S_{1/30} and Ni@C-S_{1/25} catalysts.

We further quantified the surface Ni atoms of Ni@C-S_{1/30} and Ni@C-S_{1/25} catalysts by N₂O titration, and were surprised to find that the adsorption capacity of N₂O increased with increasing sulfur contents in the catalysts (Table R11). According to the Ni loadings

(Supplementary Table 1), XRD patterns (Supplementary Fig. 2) and peak areas of Ni 2p_{3/2} XPS spectra (Supplementary Fig. 6b) for Ni@C-S_x catalysts, we believe that the number of surface Ni atoms on the S-doped Ni@C-S_x catalysts and the S-free Ni@C-S₀ catalysts is similar. Therefore, we believe that in addition to the surface Ni atoms, the sulfur can also be oxidized by N₂O. Then, the XPS analysis was carried out, and the S 2p XPS spectra for Ni@C-S_{1/30} after pulsed N₂O titration confirmed that there is not only Ni-S sulfide but also more obvious sulfur oxide over the catalyst surface (Fig. R12).

In combination with the literature (*J. Alloys Compd.* **741**, 1183-1187 (2018); *J. Phys. Chem. C.* **118**, 6934–6940 (2014)) and XPS analysis results (Fig. R12), we believe that the oxidation of sulfur by N₂O does not destroy the Ni–S bond, but forms an S–O bond on top of the S atom. For this reason, the DFT calculation was then carried out, and the result also confirmed that O atoms can form stable adsorption configurations with the S atom and its neighboring Ni atom (Fig. R13). Based on these studies, we reasonably assume that each Ni atom on the surface can be oxidized by one N₂O molecule, and each S atom on the Ni surface can also be oxidized by a N₂O molecule. In addition, according to the DRIFTS results of CO adsorption (Supplementary Fig. 8) and stable configurations of *n*-butanal adsorption (Supplementary Fig. 20), we recognize that each S atom can occupy three surface Ni atoms (threefold hollow site) and the occupied surface Ni atoms can no longer become the active sites for *n*-butanal adsorption.

Therefore, the surface sulfur coverage and Ni active sites of S-doped Ni@C-S_x catalysts can be calculated by the following formula:

$$\text{Sulfur coverage (\%)} = \frac{n_{\text{Ni@C-S}_x} - n_{\text{Ni@C-S}_0}}{n_{\text{Ni@C-S}_0}} \times 100\%$$

$$\text{Ni active sites } (\mu\text{mol}\cdot\text{g}^{-1}) = n_{\text{Ni@C-S}_x} - (n_{\text{Ni@C-S}_x} - n_{\text{Ni@C-S}_0}) \times 3$$

in which the $n_{\text{Ni@C-S}_x}$ and $n_{\text{Ni@C-S}_0}$ refer to as the moles of adsorbed N₂O per gram of S-doped Ni@C-S_x catalysts and S-free Ni@C-S₀ catalysts, respectively.

As shown in Table R9, the calculated Ni active sites for Ni@C-S₀, Ni@C-S_{1/30} and Ni@C-S_{1/25} catalysts are 26.9, 15.9 and 7.8 μmol·g⁻¹, respectively. In addition, the calculated sulfur coverages for Ni@C-S₀, Ni@C-S_{1/30} and Ni@C-S_{1/25} catalysts are 0%, 13.6% and 23.7%, respectively. These results are in agreement with our *n*-butanal-TPD/MS results and typical DFT calculation models (Ni (111) surfaces with sulfur coverage of 0%, 14% and 25%). We have now included the relevant data and discussion in the revised manuscript and Supplementary Information.

Fig. R12 | High-resolution S 2p XPS spectra for the Ni@C-S_{1/30} catalyst before and after N₂O titration.

Fig. R13 | The stable adsorption configurations of the O atoms on the model of Ni-S14% and the corresponding total energy.

Table R10 | Comparison of Ni active sites over Ni@C-S₀ catalyst quantified by H₂ and N₂O titration.

Adsorbed gas	Adsorption capacity (μmol·g ⁻¹)	Ni active sites (μmol·g ⁻¹)
H ₂	1.7	3.4 ^a
N ₂ O	26.9	26.9 ^b

^aNi sites were calculated assuming H₂/Ni_{surf} = 1/2.

^bNi sites were calculated assuming N₂O/Ni_{surf} = 1/1.

Table R11 | Surface sulfur coverage and Ni active sites of the typical Ni@C-S_x catalysts.

Catalyst	N ₂ O adsorption capacity (μmol·g ⁻¹)	Sulfur coverage (%)	Ni active sites (μmol·g ⁻¹)
Ni@C-S ₀	26.9	0.0	26.9

Ni@C-S _{1/30}	30.6	13.6	15.9
Ni@C-S _{1/25}	33.3	23.7	7.8

Pulsed H₂ chemisorption: Pulsed H₂ chemisorption was performed on a Micromeritics AutoChem II. Catalyst samples (20 mg) were reduced at 450 °C for 1 h (10 °C·min⁻¹) under a flow of 10 % H₂/Ar, and then switched to an Ar flow and cooled to 50 °C. Finally, the pulsed H₂ chemisorption was carried out at 50 °C by pulsing a mixture of 10 % H₂/Ar.

Pulsed N₂O titration: Pulsed N₂O titration was performed on a Micromeritics AutoChem II. Catalyst samples (20 mg) were reduced at 450 °C for 1 h (10 °C·min⁻¹) under a flow of 10 % H₂/Ar, and then switched to an Ar flow and cooled to 90 °C. Finally, the pulsed N₂O titration were carried out at 90 °C.

Computational details: All the spin-polarization DFT calculations were conducted based on the Vienna ab initio simulation package (VASP). The electron-ion interactions were described by the Projected Augmented-Wave (PAW) potentials, while the exchange-correlation interactions were calculated by employing the Perdew–Burke–Enzerhof (PBE) functional of Generalized Gradient Approximation (GGA). The vdW-D3 method developed by Grimme was employed to describe the van der Waals interaction. The plane-wave energy cutoff was set as 400 eV. The convergence threshold was set as 1.0×10^{-4} eV in energy and 0.02 eV per Angstrom in force. A vacuum layer of at least 15 Å was adopted to avoid the periodic interactions. The Brillouin zone was modeled by gamma centered Monkhorst–Pack scheme, in which $7 \times 7 \times 7$ and $2 \times 2 \times 1$ grids were adopted for bulk and all slab models. The O atoms were added to the Ni (111) slab model with 14% sulfur coverage and carried out structure optimization. E_{DFT} is the total energy from DFT geometry optimization models.

REVIEWERS' COMMENTS

Reviewer #1 (Remarks to the Author):

I am pleased that the authors have answered the questions I raised. Some supplementary experiments have been added and some errors have been corrected. And they have significantly improved the quality of the manuscript.

Reviewer #2 (Remarks to the Author):

I have carefully evaluated the author's response. The response is completed and all my questions have been addressed. I agree to accept this manuscript.

Point-by-point response to the reviewers' comments

Reviewer #1 (Remarks to the Author):

I am pleased that the authors have answered the questions I raised. Some supplementary experiments have been added and some errors have been corrected. And they have significantly improved the quality of the manuscript.

We really appreciate the referee's positive evaluation for our work, and are grateful to the referee for his/her insightful suggestions to help us substantially improve the quality of our manuscript.

Reviewer #2 (Remarks to the Author):

I have carefully evaluated the author's response. The response is completed and all my questions have been addressed. I agree to accept this manuscript.

We really appreciate the referee's positive evaluation for our work, and are grateful to the referee for his/her insightful suggestions to help us substantially improve the quality of our manuscript.